# Is Transformer a Stochastic Parrot? A Case Study in Simple Arithmetic Task

## Abstract

Large pre-trained language models have demonstrated impressive capabilities, but there is still much to learn about how they operate. In this study, we conduct a investigation of the autoregressive transformer's ability to perform basic addition operations. Specifically, by using causal analysis we found that a few different attention heads in the middle layers control the addition carry, with each head processing carries of different lengths. Due to the lack of globality in these attention heads, the model struggles to handle long-sequence addition tasks. By performing inference intervention on mistral-7B, partial task performance can be restored, with the accuracy on 20-digit long-sequence additions from 2% to 38%. Through fine-tuning, a new mechanism branches out for handling more complex cases, yet it still faces challenges with length generalization. Our research reveals how the model performs addition, and further provides insights into the debate on whether these models are merely statistical.

## 1 INTRODUCTION

As large pre-trained language models increase in scale, they demonstrate increasingly powerful performance on an increasing number of tasks (Brown et al., 2020). But their working principle is still a black box. As the application of large models expands, we have to start to care about safety and ethical issues (Weidinger et al., 2021). On the one hand, some studies believe that the model is just a model that relies on statistics (Bender & Koller, 2020; Merrill et al., 2021). On the other hand, some studies have found that the language model internally encodes other basic world concepts (Abdou et al., 2021; Patel & Pavlick, 2021).

Addition and subtraction, despite being the simplest arithmetic operations, are still challenging tasks for current large language models (Nogueira et al., 2021). Understanding how these models perform such operations internally is highly beneficial for improving their transparency and interpretability.

We focus on the current mainstream pre-trained models and investigate their behavior on integer addition tasks. It is worth noting that our research can be validated on larger models. For example, when giving the following input to ChatGPT 4o: "answer directly without programming: 633331+266667=", the model is highly likely to respond with 900,000 or another incorrect answer starting with 9 (the correct answer is 899,998). In this paper, we will investigate why such errors occur and implement mitigation measures.

We conducted experiments on pre-trained models, including Mistral-7B (Jiang et al., 2023) and LLaMA-7B (Touvron et al., 2023). While these models demonstrate a baseline accuracy in performing integer addition, they are far from achieving precise results. Through causal analysis, we identified a subset of attention heads, primarily in the middle layers, that are responsible for encoding digit information relevant to bitwise addition. Visualizing the associated attention patterns showed a high degree of interpretability. Ablation studies further highlighted the critical role of these heads in determining the output, governing whether the model performs simple modular addition or full addition with carry. However, as the length of the carry chain increases, the information encoded in the attention head gradually loses its significance, accompanied by the rapid decline of interpretability of the attention pattern, resulting in a decrease in the accuracy of the model (See Fig 1).

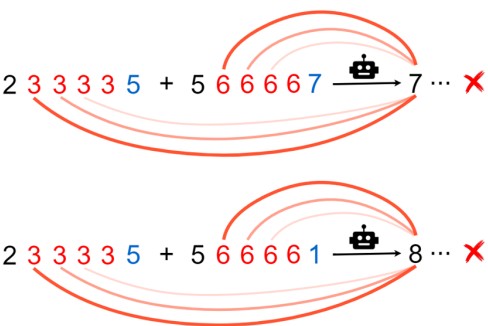

Figure 1: Two types of mistake commonly made by LLMs: the model only uses localized information for calculations, while the forward addition requires global information to handle carries. When the model cannot obtain information on whether to carry or not, it leads to incorrect outputs.

Building on our discovery of the model's underlying mechanisms, we partially restored the model's performance through inference intervention. During multiple inference runs, we manually intervened by targeting the attention heads identified through causal analysis. After each token was generated, we adjusted the attention weights by either re-weighting or ablating the attention focused on the carry positions on the last token. This intervention resulted in a significant improvement in accuracy, particularly for longer sequences.

Finally, we perform full-parameter fine-tuning on Gemma2-2B on specialized addition tasks. The results reveal that while the model refines its original mechanism for simple cases, it implements a new and more reasonable method for summation judgment to handle more complex cases. However, it continues to face challenges with length generalization.

## 2 RELATED WORK

Studying arithmetic on language models has become popular with the continuous improvement of model capabilities. Zhou et al. (2024) studies how the Transformer processes modular addition from the perspective of Fourier transforms, that the MLP layer mainly approximates the size of the results through low-frequency features, while the attention layer performs modular operations through high-frequency features. Quirke et al. (2023) shows how a one-layer model decomposes the task into parallel digit-specific computation streams and applies different algorithms to each digit by analyzing the training loss curve. Another similar work is Stolfo et al. (2023), which uses a causal mediation analysis framework to reveal the internal information flow in large language models during arithmetic reasoning tasks. Our research aims to go a step deeper and broader, providing a general framework to explain any pre-trained transformer-based autoregressive models' real internal operation process, the findings in our work could be work on a wide scale.

Lee et al. (2023); Zhou et al. (2022); Liu & Low (2023) focus on improving the performance of language models in arithmetic tasks. Their core idea is to extend the reasoning steps of the model, thereby reducing the complexity of serial calculations. This is done by employing training strategies like the "scratchpad" approach, where intermediate reasoning steps are made explicit, or by providing the model with hint prompts that guide it through mathematical expressions.

Broader interpretability research has recently focused on mechanistic interpretability (Geiger et al., 2021; Conmy et al., 2023; Wang et al., 2022). Mechanical interpretation methods, which treat the model as a computational graph composed of attention and MLP components, seek to locate the subgraph responsible for the actual task computation within the entire computational graph. Hanna et al. (2024) explores the mechanistic interpretability of how GPT-2 performs mathematical comparison (greater-than) tasks through internal circuits without explicit training.

## 3 METHOD

### 3.1 BACKGROUND

Consider two n-digit integers $X = (x_1, x_2, \ldots, x_n)$, $Y = (y_1, y_2, \ldots, y_n)$ and their addition result $Z = (z_1, z_2, \ldots, z_n)$. The numbers are tokenized digit by digit into the sequence numbers of the vocabulary, mapping to the embedding $d_i$, and the hidden state of $i$th token in the first layer is $h_i^0 = RoPE(d_i, pos_i, d_k)$, where $pos_i$ denotes the token's index $i$ in the sequence, $RoPE$ represents the rotary positional embedding operation following the definition in (Su et al., 2024) (applied in Mistral, Llama2 and Gemma2). In all our experiments, the models used the tokenizer that breaks numbers into individual digits.

Equations 1, 2, 3, and 4 outline how the transformer processes hidden states[1]. The matrices $W_Q$, $W_K$, and $W_V$ are learned linear transformations that generate the query, key, and value vectors from the input hidden states, while $W_O$ acts as the output projection matrix. $H$ denotes the number of attention heads, $d_k$ represents the hidden dimensions, and $l$ indicates the layer index.

The $\sigma$ is a non-linear function. $\gamma$ is a normalization function. $W_{up}$ and $W_{down}$ are learned weight matrices in the feed-forward network, where $W_{up}$ expands the dimensionality of the input, and $W_{down}$ reduces it back to the original dimension.

$$q_i^{(h,l)} = W_Q^{(h,l)} h_i^{(l-1)}, \quad k_i^{(h,l)} = W_K^{(h,l)} h_i^{(l-1)}, \quad v_i^{(h,l)} = W_V^{(h,l)} h_i^{(l-1)}, \tag{1}$$

$$\alpha_{ij}^{(h,l)} = \frac{\exp\left(\frac{q_i^{(h,l)} \cdot k_j^{(h,l)}}{\sqrt{d_k}}\right)}{\sum_{j' \leq i} \exp\left(\frac{q_i^{(h,l)} \cdot k_{j'}^{(h,l)}}{\sqrt{d_k}}\right)}, \quad \tilde{a}_i^{(h,l)} = \sum_{j \leq i} \alpha_{ij}^{(h,l)} v_j^{(h,l)}. \tag{2}$$

$$a_i^{(l)} = \text{Concat}\left(\tilde{a}_i^{(1,l)}, \tilde{a}_i^{(2,l)}, \ldots, \tilde{a}_i^{(H,l)}\right) W_O^{(l)}, \tag{3}$$

$$m_i^{(l)} = W_{down}^{(l)} \sigma\left(W_{up}^{(l)} \gamma\left(a_i^{(l)} + h_i^{(l-1)}\right)\right), \quad h_i^{(l)} = h_i^{(l-1)} + a_i^{(l)} + m_i^{(l)}. \tag{4}$$

Different from the way humans calculate, the autoregressive model needs to output the answer from front to back, which is more challenging. Consider the process needed for the model output to be the correct $z_i$, there are three cases. Case 1: When $x_{i+1} + y_{i+1} < 9$, $z_i = (x_i + y_i) \mod 10$. Case 2: When $x_{i+1} + y_{i+1} > 9$, then $z_i = (x_i + y_i) \mod 10 + 1$. Case 3: When $x_{i+1} + y_{i+1} = 9$, we need to check if there is a carry from subsequent digits. If $x_{i+2} + y_{i+2} > 9$, then $z_i = (x_i + y_i) \mod 10 + 1$; if $x_{i+2} + y_{i+2} < 9$, then $z_i = (x_i + y_i) \mod 10$; if $x_{i+2} + y_{i+2} = 9$, continue checking further.

For simplicity, we refer to an equation with a carry chain of length $d$ as *CCd* (Carry Chain). For example, 44 + 28 is *CC1*, 356 + 247 is *CC2*, and 35556 + 24447 is *CC4*. On the other hand, an equation that only has the chain format without actual carry is called *OCd* (Only Chain). For example, 345 + 252 is *OC2*, and 46612 + 33385 is *OC4*.

We analyze the behavior of Mistral-7B on $OCd$ and $CCd$ tasks (Figure 2), where $d$ ranges from 1 to 15. Each $d$ corresponds to 200 samples (100 $OC$ and 100 $CC$). For $OC$ inputs $X + Y$ and $CC$ inputs $X' + Y'$, we compute the average probability distributions $p(z_1|X, Y)$, $p(z_1 + 1|X, Y)$, $p(z_1'|X', Y')$, and $p(z_1' - 1|X', Y')$. To simulate realistic scenarios, a randomly generated sequence of length $d$ is appended to each input pair, resulting in a total length of $2d$.

The results show that, in addition to the overall probability exhibiting a decreasing trend as the sequence length increases, once $d > 1$, the average probability of errors $p(z_1' - 1|X', Y')$ on the CC task exceeds that of correct probabilities $p(z_1'|X', Y')$. Furthermore, when $d > 8$, the lines nearly overlap, indicating that the model can hardly distinguish *CC* tasks from *OC* tasks.

---

[1]For simplicity, we omit details of positional encoding in each layer, the implementation of the mixture of experts in Mistral, and the grouped query attention mechanism.

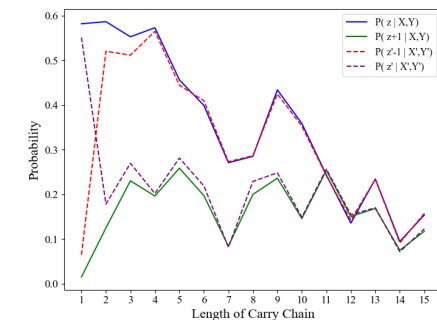

Figure 2: The average probability output of Mistral-7B on *OC* and *CC* tasks, with each value of length corresponding to 200 samples.

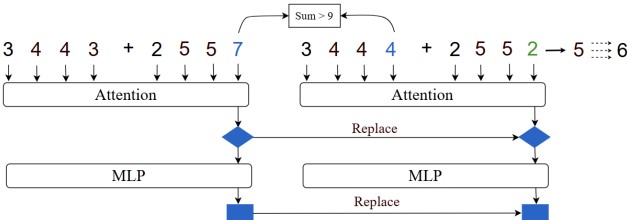

Figure 3: An interpretation of the causal analysis is presented in this example, where $y'_{d+1} + x_{d+1} > 9$, causing the output shift from 5 to 6.

## 3.2 CAUSAL ANALYSIS

Causal analysis (Vig et al., 2020; Pearl, 2022; Meng et al., 2022) is a technique based on activation replacement, helping to reveal the causal roles of internal model components and understand how they contribute to outcomes. We first create a set of two similar but different inputs, *OCd* input $X + Y$ and *CCd* input $X' + Y'$. Naturally, $z_1 = x_1 + y_1$, $z'_1 = x'_1 + y'_1 + 1$. And set restriction: $x_i = x'_i, y_i = y'_i (i < d+1)$; $y'_{d+1} + x_{d+1} > 9$ or $x'_{d+1} + y_{d+1} > 9$ (randomly chosen). Figure 3 explains how causal analysis can impact model output. Two *OC* inputs can produce the same results as long as the constraints are maintained (see Appendix A).

We conduct three rounds of model inference. The symbols below refer to equations 1, 2, 3, 4.

- **In the first run:** $(X + Y)$ as the input to obtain the final probability output $p(z_1|X, Y)$ and $p(z'_1|X, Y)$ and collect the activation $o$ at $y'_{d+1}$ token position if $y'_{d+1} + x_{d+1} > 9$, or $x'_{d+1}$ token position if $x'_{d+1} + y_{d+1} > 9$, $o \in \{m^{(1)}_{d+1}, \ldots, m^{(L)}_{d+1}, v^{(1)}_{d+1}, \ldots, v^{(L)}_{d+1}\}$.

- **In the second run:** $(X' + Y')$ as the input and collect the activation $o'$ at $y'_{d+1}$ token position if $y'_{d+1} + x_{d+1} > 9$, or $x'_{d+1}$ token position if $x'_{d+1} + y_{d+1} > 9$, $o' \in \{m'^{(1)}_{d+1}, \ldots, m'^{(L)}_{d+1}, v'^{(1)}_{d+1}, \ldots, v'^{(L)}_{d+1}\}$.

- **In the third run:** $(X + Y)$ as the input and replace the activation $o$ with $o'$ to obtain the probability output $p^*(z'_1|X, Y)$.

When intervening in reasoning, we sequentially use $o'$ to override the original activation $o$ to change the model's probability output. Intuitively, this should lead to an increase in the model's output probability for $z'$. The total effect is defined as 5. We used 100 sets of number pairs as input for Mistral-7B and calculated their average $TE$. For activation replacement of the attention layer, $v$ is chosen instead of $\tilde{a}$ or $a$ because non-trivial result first and only occurs on $v$. Causal analysis of other components ($\tilde{a}$ and $a$) refer to Appendix A.

$$\text{Total Effect (TE)} = p^*(z'_1|X, Y) - p(z'_1|X, Y) \tag{5}$$

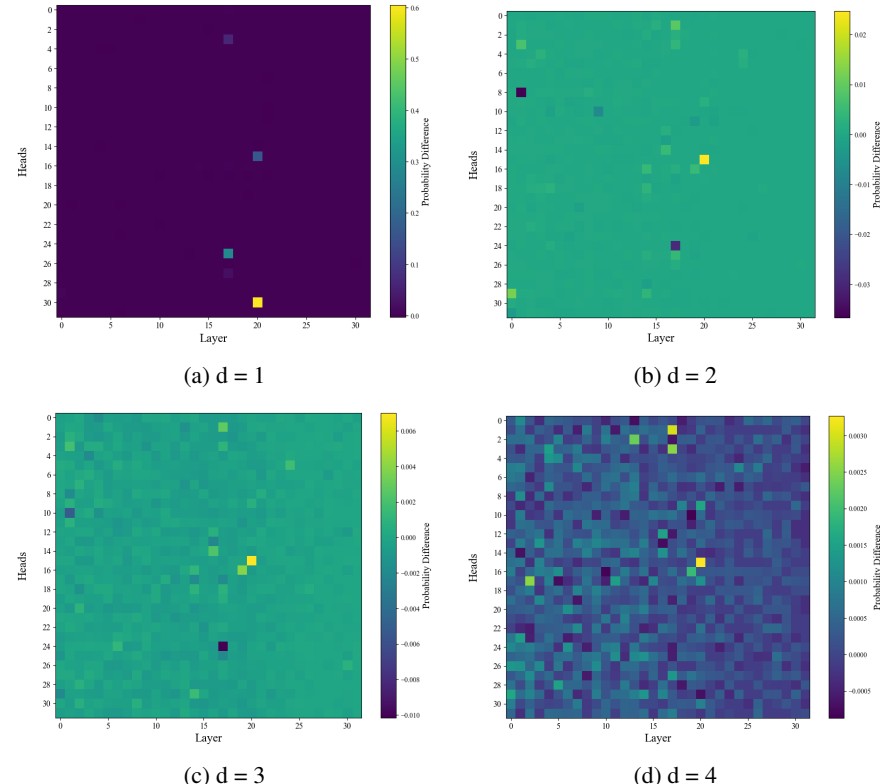

(a) d = 1

(b) d = 2

(c) d = 3

(d) d = 4

Figure 4: The attention heads located through causal analysis show that for mistral-7B under different values of $d$. Mistral-7B has 32 attention heads per layer, for a total of 32 layers.

Through our experimental analysis, we have the following findings (See Fig 4): the most significant impact occurs in the middle to later layers, specifically between layers 15 and 20. When $d = 1$, a few attention heads exhibit a strong influence on the output, leading to up to a 50% difference in probability. However, as $d$ increases, the maximum probability difference declines rapidly, with only 2% difference at $d = 2$. Some attention heads, such as (Head 15, Layer 20), show a broader but much weaker range of influence. We use TSNE (Van der Maaten & Hinton, 2008)) to visualize internal shift within the model, where the hidden states for the $OC1$ and $CC1$ became distinctly separated after the attention layers, as shown in Appendix B. MLP impacts the results by mostly contributing to the located attention heads. (Causal analysis of MLP, other models, and effect breakdown, please refer to Appendix A).

### 3.3 ATTENTION IMPLEMENTS INCOMPLETE CARRY

The equation 2 illustrates how the replaced $v$ vector affects the output through the attention pattern. To recall that the $v$ vector position we replaced in section 2 is $y'_{d+1}$ (or $x'_{d+1}$ depending on the sample), so $i = y'_{d+1}$, and the replaced $v$ affects the output through the attention weight $\alpha_{n,y'_{d+1}}$, where $n$ the last sequence position.

Among the attention heads observed in the causal analysis, we visualize the top two attention heads that cause the largest $TE$ for each value of $d$ (See Fig 5).

It is observed that when the "=" token appears, the model focuses on $x_{i+1}$ and $y_{i+1}$ in order to calculate $z_i$, just like a double pointer. As the length of the output sequence increases, it continues to move towards, forming a double staircase pattern. However, when $d > 1$, the attention head responsible for moving the digit information loses most of this staircase pattern. Instead, the attention weights are unevenly distributed on each digit. The formation of the attention pattern is independent of whether the input itself contains carry.

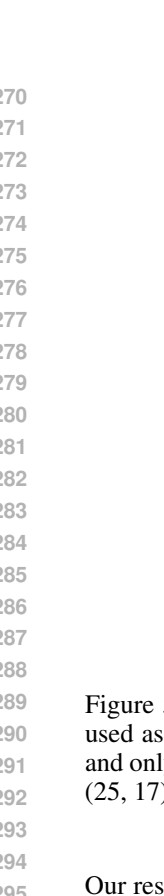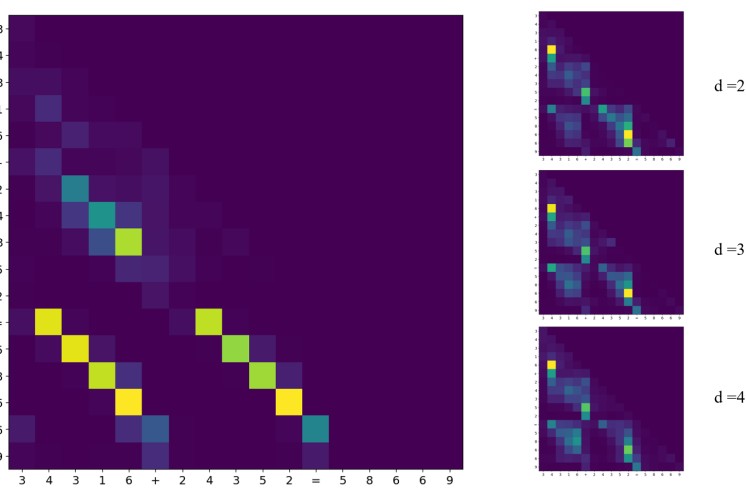

d = 1

Figure 5: The result of superimposing the attention heads on Mistral-7B, 34316+24352=58669 is used as a demonstration example. For simplicity, the attention pattern displayed omits the prompt and only retains the question. For $d$ from 1 to 4, the selected top 2 attention heads (Head, Layer) are (25, 17) and (30, 20); (15, 20) and (3, 17); (15, 20) and (16, 19); (15, 20) and (1, 17)

Our research focuses on two objectives: first, to assess the influence of the targeted attention heads on the corresponding $CC$ task, the second is to investigate whether the model's significant performance decline on $CC$ tasks with $d > 1$ is solely due to insufficient attention weights allocation. To address the first objective, we perform ablation on the top two attention heads for the corresponding $CC$ task, by setting $\alpha_{n,y'_{d+1}} = 0, \alpha_{n,x'_{d+1}} = 0$ to eliminate the influence of the $v$ vector. Additionally, we randomly select attention heads in the same layer and same token position as the ablation heads for comparison. For the second question, we dynamically perform re-weighting (6) according to the input.

$$\alpha_{n,x'_{d+1}}, \alpha_{n,y'_{d+1}} = \alpha_{n,x'_{d+1}} + \lambda, \alpha_{n,y'_{d+1}} + \lambda \tag{6}$$

Table 1: Ablation study (with $\lambda = 0.6$) on Mistral-7B, Llama2-7B, and Gemma-7B models. Numbers in parentheses are the baseline for the $OC$ task.

| Model | Method | Task | | | |
|---|---|---|---|---|---|
| | | $CC1$ | $CC4$ | $CC6$ | $CC10$ |
| Mistral-7B | Baseline | 99.21(98.32) | 29.99(80.24) | 20.31(67.18) | 17.93(23.26) |
| | Zero ablation | **34.80** | 24.51 | 19.21 | 17.09 |
| | Random ablation | 96.99 | 29.46 | 21.32 | 17.45 |
| | Re-weighting | 98.12 | **41.83** | **23.87** | **19.20** |
| Llama2-7B | Baseline | 91.17(98.09) | 48.21(44.23) | 17.33(12.41) | 0.67(0.21) |
| | Zero ablation | **17.51** | 42.14 | 17.23 | 0.67 |
| | Random ablation | 88.92 | 49.21 | 17.30 | 0.68 |
| | Re-weighting | 89.32 | **58.43** | **22.78** | **4.63** |
| Gemma-7B | Baseline | 99.33(99.21) | 80.60(37.26) | 49.98(31.65) | 25.17(26.88) |
| | Zero ablation | **1.47** | 49.72 | 28.69 | 20.64 |
| | Random ablation | 96.84 | 78.53 | 48.57 | 25.07 |
| | Re-weighting | 95.10 | **92.73** | **58.48** | **31.46** |

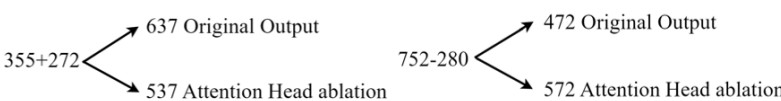

Figure 6: Zero ablation causes the output of the model to revert back to modular addition, and the same applies to subtraction.

For the $CC1$ task with high accuracy, there is a strong correlation between attention and model output (see Table 1). Ablating the top two attention heads has an immediate and dramatic effect. Notably, after ablating on Gemma, the accuracy of the $CC1$ task drops to just 1.47%. The ablation removes the carry and reverts the output of the first digit to $z_1' - 1$ (See Fig 6). As $d$ increases, the impact of ablating attention heads diminishes, and accuracy declines accordingly.

When re-weighting weights to match $CC1$ level, we observed varying degrees of improvement across different $CCd$ tasks. However, this improvement was still limited and did not restore accuracy to the $CC1$ level. Additionally, as $d$ increases, the effect of re-weighting continues to diminish. This suggests that the model's difficulty in handling carries is not solely due to insufficient attention weight allocation, but rather a deeper issue in the model's mechanism for processing such operations.

Interestingly, the three models exhibit distinct patterns in handling tasks, and when uncertain about the presence of a carry, they tend to default to one of these patterns. Mistral-7B achieves higher accuracy on $OC$ tasks compared to $CC$ tasks, while Gemma shows the opposite, with better performance on $CC$ tasks than $OC$ tasks. Llama2 falls between these two models. The relatively low accuracy of Llama2 can be attributed to its inherently lower performance in modular addition compared to the other two models (see Appendix C)..

To summarize the experimental findings in this section, pre-trained autoregressive models rely on the staircase attention patterns to transmit carry-encoded value tokens to the final token for prediction. When this information transmission is disrupted, the model defaults to modular addition. As the carry length increases, the model loses its ability to transmit this information, leading to disordered attention patterns and a loss of carry-related information in the value vectors.

## 4 INFERENCE INTERVENTION

Based on the findings in section 3, we performed a full intervention enhancement experiment on the model's addition task. Many of the model's errors stemmed from incorrect handling of carries—either missing a carry or introducing an unnecessary one. In contrast, errors related to modular addition were relatively minor (See Appendix B). The primary goal of this experiment was to explore the potential for improving the model's performance without additional training, by attempting to restore its internal mechanisms, rather than transforming the model into a fully accurate calculator.

### 4.1 EXPERIMENTS

Our experiment follows the following steps: when the model processes $X + Y$, it dynamically intervenes in internal activation. When the model generates $z_i$, we determine whether there is a carry from the following sequence. We melt the model's attention weights for $x_{i+1}$ and $y_{i+1}$ if there is no carry and let the model do modular addition $(x_{i+1} + y_{i+1}) \bmod 10$. If there is a carry from a carry chain of length $m$, the attention weight $a_{n,i+m}$ is re-weighted. Here, the attention heads are selected from the top 2 attention heads that cause the maximum $TE$ when $d = m$. For a more detailed algorithm explanation please refer to Appendix C.

In the experiment, the number pairs corresponding to each length $n$ were randomly sampled from $10^{n-1}$ to $10^n$, and each length contained 9000 questions. The model used greedy sampling. Due to the need for $n$ times of inference interferences for numbers with a length of $n$, this computationally intensive experiment ends at a length of 20.

Similar to our experiment, activation intervention is a technology that adjusts the activation of some components in the reasoning process to modify the behavior of the model. This also stems from the work on the interpretability of deep neural network mechanisms (Adi et al. (2016); Finlayson et al. (2021); Vig et al. (2020)), but our experiments generally only involve modifications to dozens of scalars.

---

**Algorithm 1** Model Inference with Ablation and Re-weighting

---

1: **Input:** $X + Y$. $L$: The last position in the sequence. $m$: Length of the carry chain. $A_m$: Top 2 attention heads located in causal analysis when $d = m$.
2: **Output:** $Z$
3: **for** $i \leftarrow 0$ **to** $n - 1$ **do**
4:   **if** $x_{i+1} + y_{i+1} < 9$ **then**
5:     Perform inference: $z_i \leftarrow \text{Model}(X, Y)$ with ablation $(a_{L,x_{i+1}}, a_{L,y_{i+1}})$ on Head $A_1$
6:     Output: $z_i$
7:   **else**
8:     **if** Carry exist: **then**
9:       Perform inference: $z_i \leftarrow \text{Model}(X, Y)$ with re-weighting $(a_{L,x_{i+m}}, a_{L,y_{i+m}})$ on Head $A_m$
10:       Output: $z_i$
11:     **else**
12:       Perform inference: $z_i \leftarrow \text{Model}(X, Y)$ with ablation $(a_{L,x_{i+1}}, a_{L,y_{i+1}})$ on Head $A_1$
13:       Output: $z_i$
14:     **end if**
15:   **end if**
16:   $X + Y \leftarrow X + Y + z_i$
17: **end for**

---

## 4.2 RESULTS

The result (See Fig 7) shows improvement in most cases, with longer sequences experiencing more significant improvements. The lack of improvement for sequences shorter than length 5 is that the proportion of these randomly sampled short sequences that contain $CC2$ or above is relatively low. Specifically, for sequences of length 3, only about 17% of the questions contain $CCd$ where $d \neq 1$, this proportion increases to 26% at length 5, 56% at length 10, and 86% at length 20. Table 1 shows that the improvement brought by re-weighting is mainly in more difficult cases, but harms the case of $CC1$, where the model already performs sufficiently well.

Additionally, this improvement has an upper limit, as accuracy remains close to zero for sequences of length 60 (shown in Appendix C). This is partly due to the limitations of attention discussed in Section 3, where simply adjusting weights cannot fully compensate for performance decline caused by the information loss. Another critical factor is that while intervention improves the model's handling of carries, it does not enhance the model's accuracy in performing basic modular addition. Additionally, many errors arise from number misalignment, where the model incorrectly adds $x_i$ with $y_j$ where $i \neq j$ (See Appendix B). Many studies (McLeish et al., 2024; Shen et al., 2023) have that positional encoding is a primary cause of this issue.

## 5 FINE-TUNING

Furthermore, we extend our investigation to the model's internal processing after fine-tuning on more complex tasks, conducting full-parameter fine-tuning experiments on the Gemma2-2B model.

The dataset includes questions of $CCd$, $OCd$, and randomly generated number pairs. To ensure that modular addition does not affect the results, 40% of the dataset consists of randomly sampled numbers with a length upper limit of 80, 30% consists of $CCd$ tasks, and 30% consists of $OCd$ tasks, where $d$ represents the maximum training length of the carry chain. The fine-tuned model results are shown in Figure 8. The detailed training parameters are provided in Appendix E.

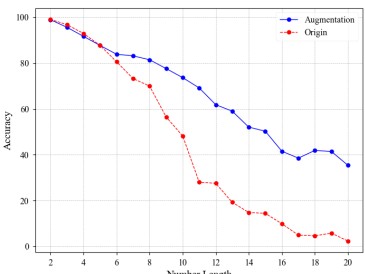

Figure 7: Comparison of accuracy between the baseline and inference intervention augmentation on Mistral-7B.

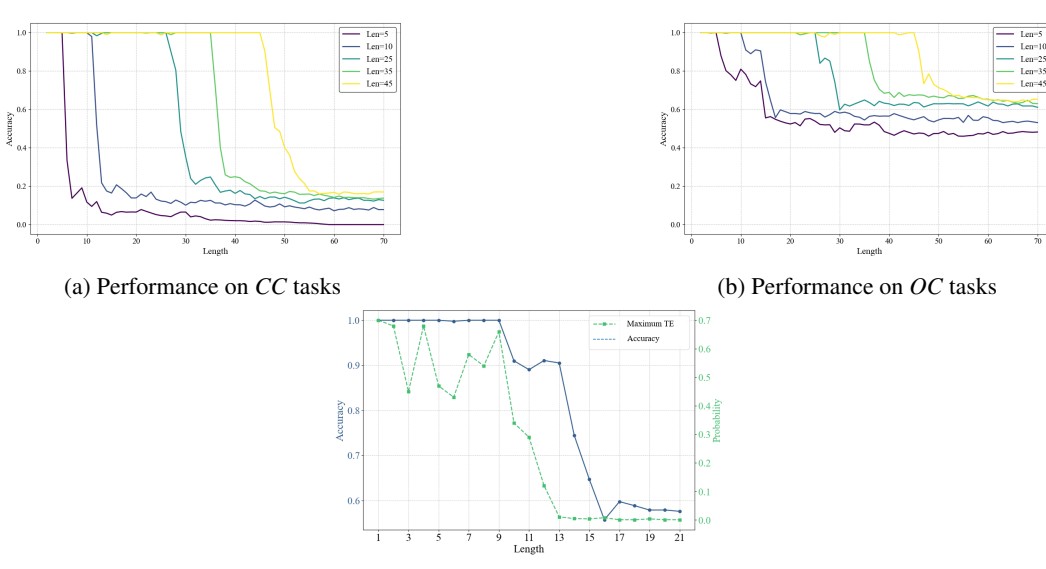

(a) Performance on $CC$ tasks

(b) Performance on $OC$ tasks

(c) Accuracy on $CC$ tasks and maximum $TE$ on the model trained on a maximum training length of 10.

Figure 8: (a), (b): Performance of Gemma2-2B on $CCd$ and $OCd$ tasks after fine-tuning, legend represents the max training length $d$. (c): The accuracy of $CCd$ task on model training on training length $d = 10$, along with the corresponding maximum $TE$ tracked tracked throughout.

## 5.1 GENERALIZATION

Related studies (Sabbaghi et al., 2024; McLeish et al., 2024; Kazemnejad et al., 2024) have highlighted the critical role of positional encoding in enabling length generalization for arithmetic tasks. Since arithmetic relies fundamentally on the alignment of digits, positional encoding that effectively captures this structural information greatly enhances generalization (Sabbaghi et al., 2024; McLeish et al., 2024). However, handling long carry chains remains a significant challenge for models (Sabbaghi et al., 2024), as it requires not only precise digit alignment but also conditional reasoning based on the sum of individual digits. While Gemma2-2B employs RoPE, study (Kazemnejad et al., 2024) indicates that it still faces limitations in achieving length generalization.

Figure 8 shows the fine-tuning performance of Gemma2-2B. The model seems difficult to achieve noticeable generalization. The accuracy drops sharply after surpassing the maximum training length. We attempt to explain this phenomenon from the perspective of attention head formation by measuring the maximum $TE$ that the attention heads can cause. The results (8c) indicate that beyond the training length, model performance declines in tandem with maximum $TE$, suggesting a lack of attention heads capable of effectively transferring carry digit information. The detailed causal analysis heat maps are visualized in Appendix D.

## 5.2 EMERGENCE OF NEW PATTERN

We perform a causal analysis on the fine-tuned model with a maximum training length $d = 10$. Pre-trained models handle cases like $OC1$ and $CC1$ by allocating attention to the digit immediately following the current calculation digit. After fine-tuning, the model retains and refines the original mechanism (see Appendix D) while also developing a new strategy (see Figure 9) to address more complex cases ($d > 2$). The original mechanism continues to handle simple cases($d \leq 2$) in parallel with the new strategy. Notably, the model introduces a specialized attention pattern, referred to as the Target Head, which focuses directly on the actual carry digit (a centralized attention weight on digit "7" in Figure 9). Simultaneously, another attention mechanism, named the Detection Head, emerges. This head becomes active even before the "=" token appears, transmitting the corresponding $x_i$ information to the current token $y_i$ token when it appears.

A guess is that the detection head obtains the information of $x_i$ and $y_i$ to determine whether $x_i + y_i$ is greater than 9. The carry chain length-related information is then passed to the Target Head, enabling it to focus on the actual carry token and execute the carry operation. To validate this, we performed zero ablation on the Detection Head and observed a disruption in the Target Head's attention patterns, resulting in a significant accuracy decrease for complex cases, while simpler cases remained unaffected (see ablation studies in Appendix D). Intuitively, the differentiation of attention functions is more reasonable, since it involves the concept of sum judgment, but it is still unclear why the model does not continue to use the original mechanism and how the information is delivered. Addressing this question will require further studies.

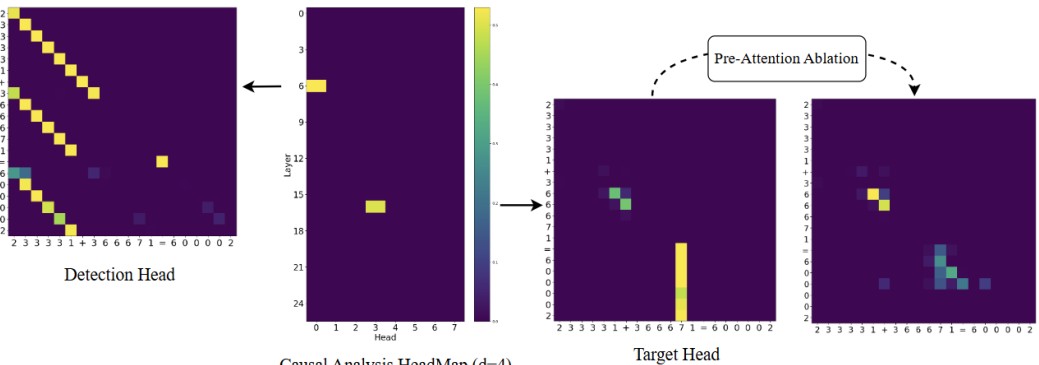

Figure 9: Causal Analysis on fine-tuned Gemma2-2B. "233331+366671=600002" is used as the demonstration example. Left: The Detection Head Pattern. Middle Left: Causal analysis on all layers and heads. Middle Right: Target head patterns. Right: Target head patterns with detection head melted.

## 6   CONCLUSIONS

In this study, we investigated how pre-trained autoregressive language models perform addition operations.

We found that the model relies on localized attention distribution for handling carry operations, which makes it challenging for the model to process inputs with long sequences. We attempted to restore the model's task performance by intervening in attention during inference without additional training. While some task performance was recovered, the model's inherent limitations remained. Finally, fine-tuning on specialized addition tasks revealed that the model develops a new and more efficient mechanism to handle complex cases, while retaining the original mechanism for simpler situations, yet it continues to struggle with length generalization. This loss of generalization ability is related to the loss of ability to form functional attention heads.

Our findings offer valuable insights into how language models process arithmetic tasks and serve as a reference point for evaluating whether current language models rely solely on statistical patterns rather than deeper reasoning mechanisms.

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

## A  APPENDIX - ADDITIONAL INFORMATION FOR CASUAL TRACING

This section discusses causal analysis for other models and some interactions we observed between attention and MLP.

The attention head located in Llama2-7B is further forward than Mistral (See Fig 10), and significant $TE$ occurs earliest in the 13th layer. Similarly, as $d$ increases, the maximum $TE$ rapidly decreases.

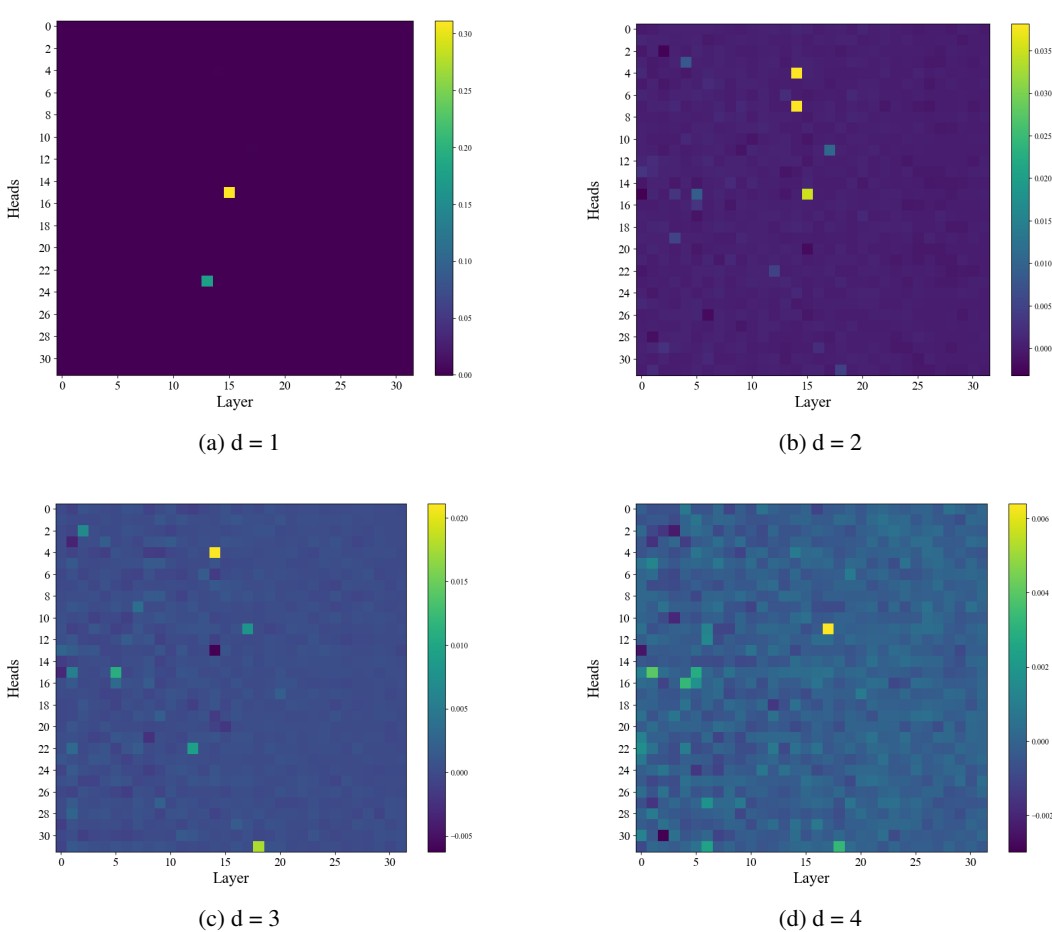

(a) d = 1

(b) d = 2

(c) d = 3

(d) d = 4

Figure 10: The attention heads located through causal analysis show that for Llama2-7B under different values of $d$. Llama2-7B has 32 attention heads per layer, for a total of 32 layers.

Figure 11 shows the causal analysis results based on two $OCd$ inputs. The targeted heads are similar to the heads in Figure 4.

When performing activation intervention on a certain component, its total effect can be divided into two parts (Vig et al., 2020): one is that the component directly affects the output probability by writing the residual stream value to cause direct effects ($DE$), and the other is that the residual stream passes the influence to downstream components to cause indirect effects ($IE$) (See Figure 13).

We found that the $TE$ caused by MLP mainly comes from the indirect effect on downstream attention components, especially on the Top 2 attention heads. We set up an additional experimental process to distinguish the degree of influence between the two. Specifically, when we perform activation intervention on $m_1'^l$ we fix the top 2 attention heads components as their original activation $a_i^{Top2}$.

The results (See Fig 14) show that the impact of MLP on the results was concentrated in the early stage of the model, and most of it was earlier than the influence range of the attention heads. After

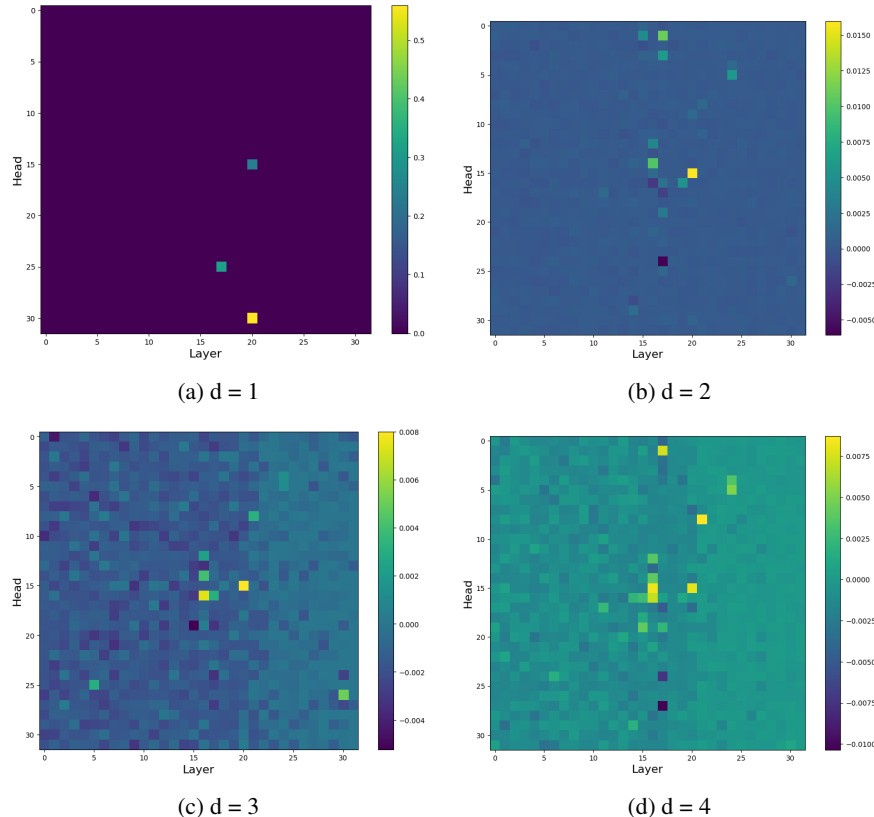

(a) d = 1

(b) d = 2

(c) d = 3

(d) d = 4

Figure 11: The attention heads located through causal analysis based on two $OC$ inputs in Mistral-7B

restoring the top 2 attention heads, the impact of MLP decreased to an insignificant level. This may represent that the role of MLP in the early stage is to provide the basic work of information processing for attention layers.

In addition to the activation replacement of $v$, we also conducted experiments on $\tilde{a}$ and $a$. (no concept of attention head in $a$.) The results (See Figure 15b, 15a, 16b, 16a) show that these components are difficult to significantly impact the output probability (even if $d = 1$).

# B    APPENDIX - VISUALIZATION

We provide some visualizations in this section, mainly including the process of handling modular additions and additions with carry inside the model.

In Figure 17, it is observed that in layer 12, similar labels (also of similar colors) are mixed together. At layer 13, the clustering pattern immediately undergoes a sudden change, and the model distinguishes between carry and non-carry equations, which are much closer to the output of the final layer. Layer 13 is also the earliest layer to be located through causal analysis (See Fig 10a).

Similarly, a similar process also occurs in the mistral-7B model (See Fig 18). After passing through the 17th layers, the model quickly distinguishes between $CC$ and $OC$ tasks, the causal analysis location of the 17th layer refers to 4a.

For modular addition, the visualization results show that the model is implemented in a progressive manner, rather than dealing with abrupt changes like handling carry.

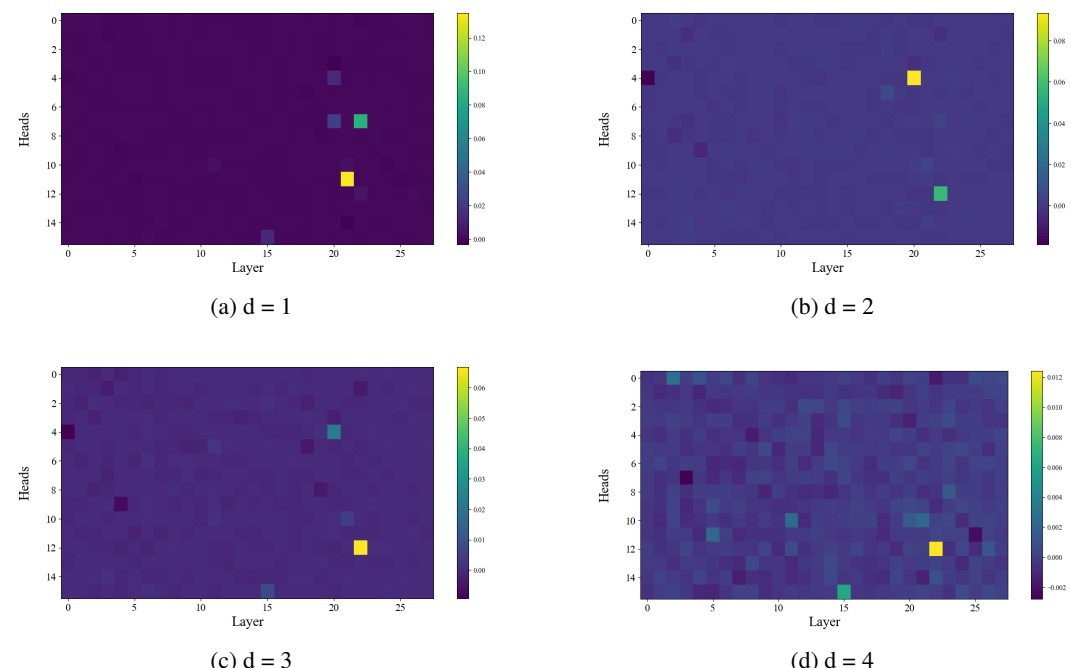

(a) d = 1      (b) d = 2

(c) d = 3      (d) d = 4

Figure 12: The attention heads located through causal analysis show that for Gemma-7B under different values of $d$. Gemma-7B has 16 attention heads per layer, for a total of 26 layers.

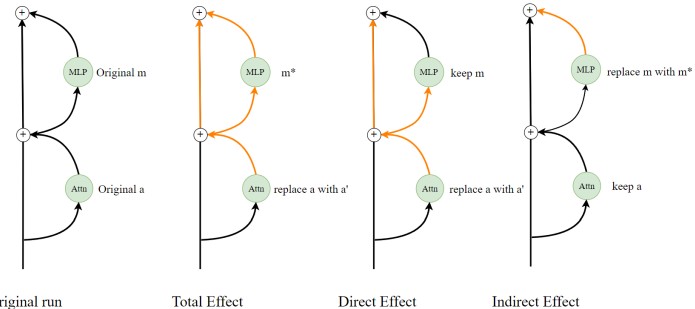

Figure 13: Total Effect, Direct Effect and Indirect Effect

## C APPENDIX - DETAILS IN INFERENCE INTERVENTION

In this section, we give more details about the inference intervention.

We first classify the incorrect answer obtained by the model into four situations. When the model treats the $CC$ question as the $OC$ question, it is called a missing carry; When treating the $OC$ question as a $CC$ question, it is called extra carry; alignment errors caused by incorrect numerical alignment; basic modular addition errors.

The model runs on a dataset of randomly generated numbers of a specified length. By classifying each error, it can be found that the most common error made by the model in low sequence lengths is missing carry (See Fig 19). As the length increases, errors caused by alignment account for the vast majority, rather than basic modular additions. Our inference intervention is only optimized for missing carry and extra carry situations.

We give a more detailed explanation of the algorithm for inference intervention (See Algorithm2). The algorithm explains more concretely about the carry detection progress.

**Algorithm 2** Model Inference with Ablation and Reweighting

1: **Input:** $X + Y$. $L$: The last position in the sequence. $m$: Length of the carry chain. $A_m$: Top 2 attention heads when $d = m$ in causal analysis.
2: **Output:** $Z$
3: **for** $i \leftarrow 0$ **to** $n - 1$ **do**
4:     Initialize: $m \leftarrow i$
5:     **if** $x_{i+1} + y_{i+1} < 9$ **then**
6:         Set ablation: $a_{L,x_{i+1}}, a_{L,y_{i+1}} \leftarrow 0$
7:         Perform model inference: $z_i \leftarrow \text{ModelInference}(X, Y, \text{ablation}(a_{L,x_{i+1}}, a_{L,y_{i+1}}))$
8:         Output: $z_i$
9:     **else if** $x_{i+1} + y_{i+1} > 9$ **then**
10:         Re-weighting: $a_{L,x_{i+1}}, a_{L,y_{i+1}} \leftarrow A_m$
11:         Perform model inference: $z_i \leftarrow \text{ModelInference}(X, Y, \text{reweighting}(a_{L,x_{i+1}}, a_{L,y_{i+1}}))$
12:         Output: $z_i$
13:     **else**
14:         **while** $x_{i+1} + y_{i+1} == 9$ **do**
15:             $m \leftarrow m + 1$
16:             **if** $x_m + y_m < 9$ **then**
17:                 Set ablation: $a_{L,x_m}, a_{L,y_m} \leftarrow 0$
18:                 Perform model inference: $z_i \leftarrow \text{ModelInference}(X, Y, \text{ablation}(a_{L,x_m}, a_{L,y_m}))$
19:                 Output: $z_i$
20:             **else if** $x_m + y_m > 9$ **then**
21:                 Re-weighting: $a_{L,x_m}, a_{L,y_m} \leftarrow A_m$
22:                 Perform model inference: $z_i \leftarrow \text{ModelInference}(X, Y, \text{reweighting}(a_{L,x_m}, a_{L,y_m}))$
23:                 Output: $z_i$
24:             **end if**
25:         **end while**
26:         Set ablation: $a_{L,x_{i+1}}, a_{L,y_{i+1}} \leftarrow 0$
27:         Perform model inference: $z_i \leftarrow \text{ModelInference}(X, Y, \text{ablation}(a_{L,x_m}, a_{L,y_m}))$
28:         Output: $z_i$
29:     **end if**
30:     $X + Y \leftarrow X + Y + z_i$
31: **end for**

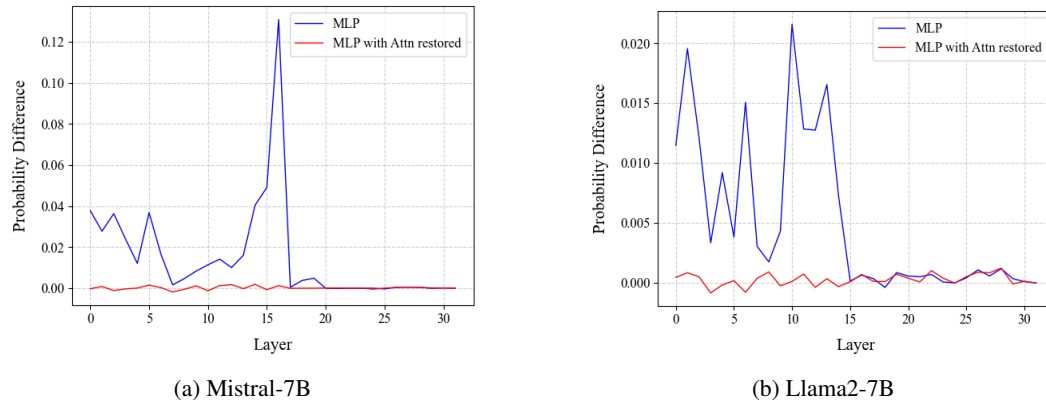

(a) Mistral-7B

(b) Llama2-7B

Figure 14: The *TE* caused by MLP activation replacement and the effect caused by restoring the top 2 attention heads, the results are averaged among 100 samples.

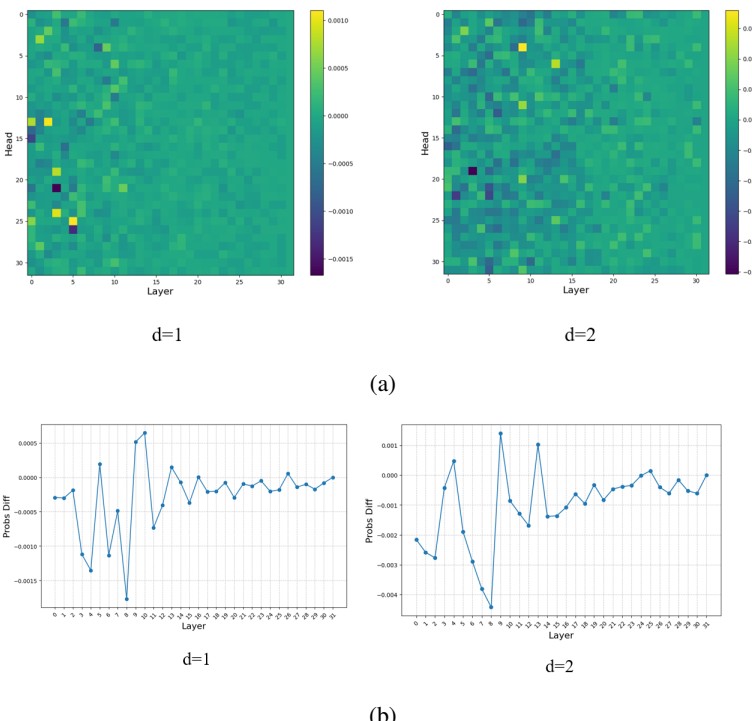

d=1

d=2

(a)

d=1

d=2

(b)

Figure 15: Casual analysis of other components in Llama2-7B. (a) Replacing $\tilde{a}$. (b) Replacing $a$.

The improved accuracy data beyond length of 20 is listed in table 2.

# D  APPENDIX - DETAILS IN FINE-TUNING

We first perform a causal analysis on the fine-tuned model (trained on a maximum length of 10) under different $d$ values (See Figure 20).  where the location of attention heads of d=1 case is different from others, representing a differentiation of functional attention heads.

In section 5.2, we discussed the emergence of a new pattern, the original mechanism remains and is refined. Figure 21 shows the top-2 attention heads overlay pattern located in casual analysis.

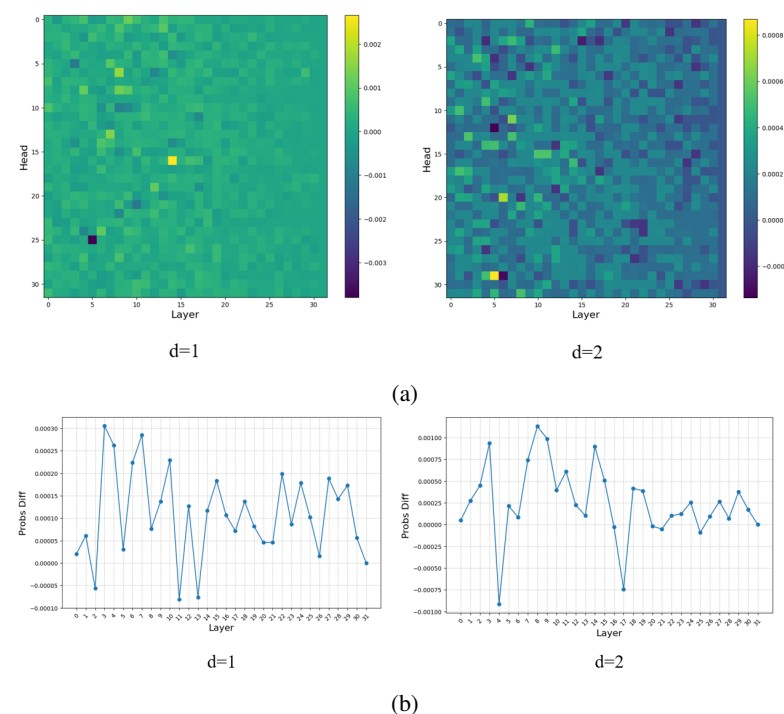

Figure 16: Casual analysis of other components in Mistral-7B. (a) Replacing $\tilde{a}$. (b) Replacing $a$.

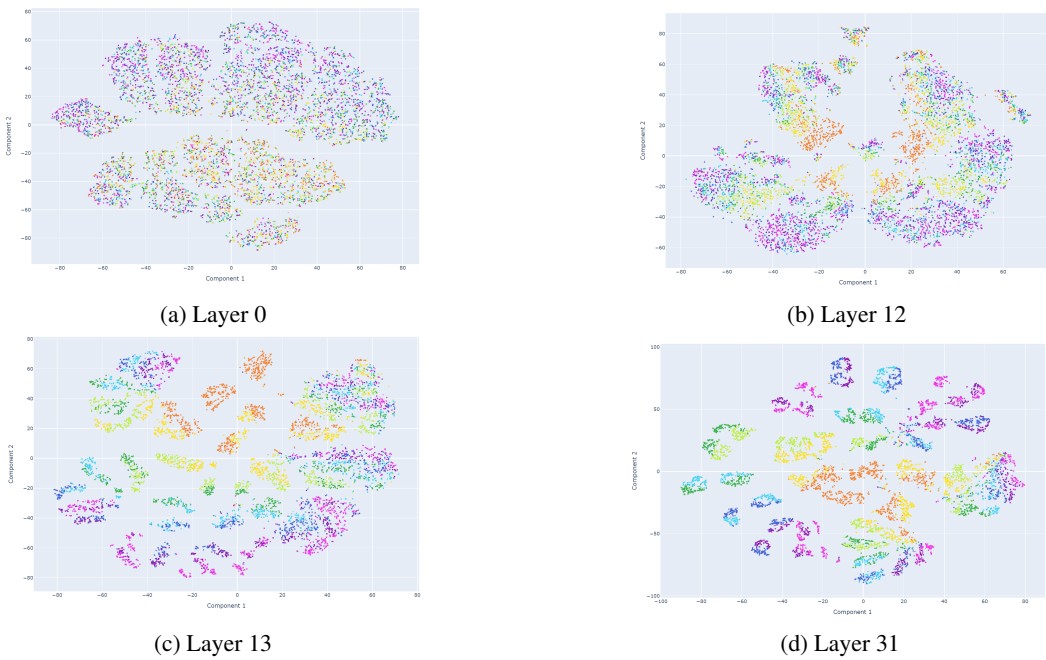

Figure 17: TSNE visualization of the last token hidden state on Llama2-7B, each color represents the label $z_1$.

In terms of the new mechanism of detection heads and target heads, to quantify their importance, we perform an ablation study on them (See table 3).

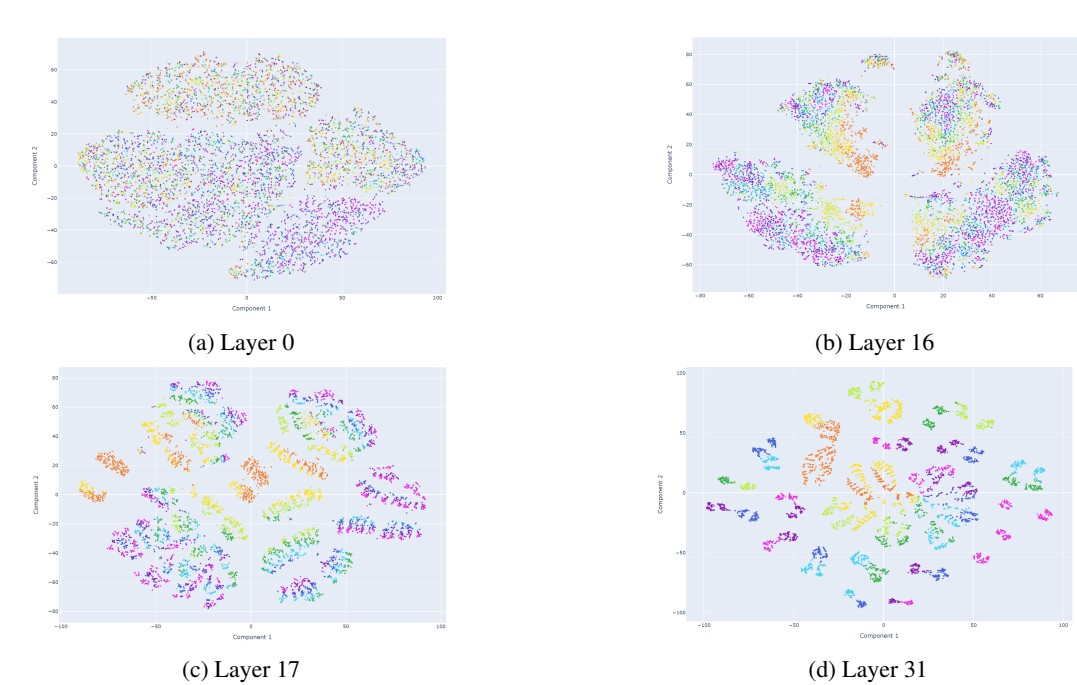

Figure 18: TSNE visualization of the last token hidden state on Mistral-7B, each color represents the label $z_1$.

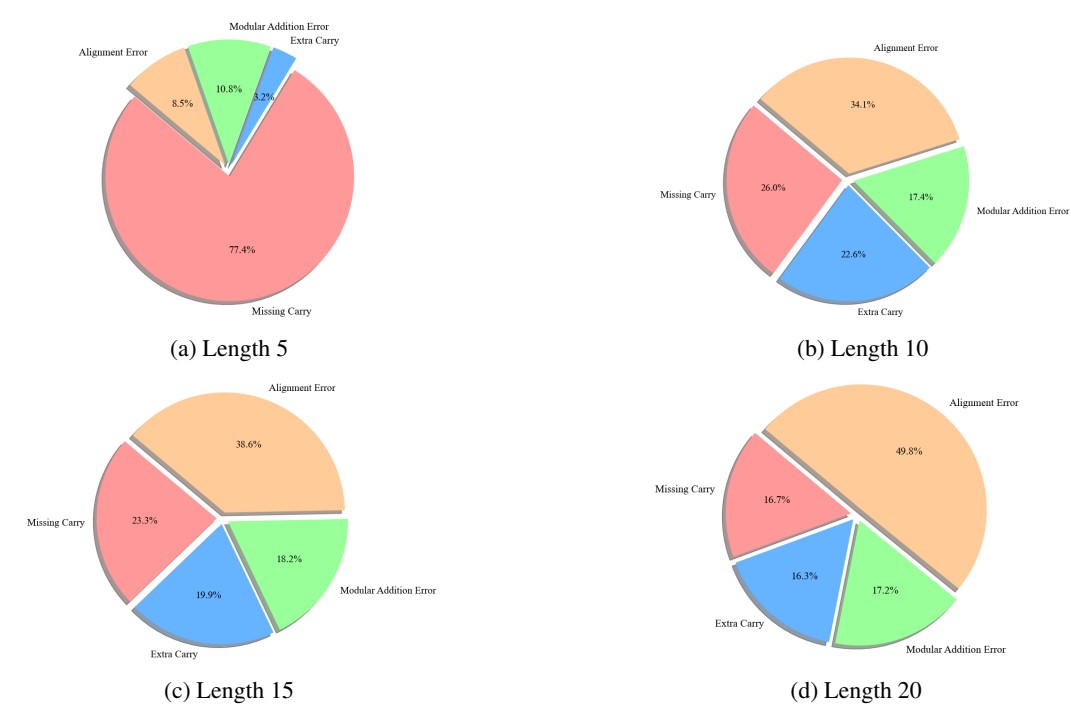

Figure 19: Four types of error that Mistral-7B makes on different lengths.

Table 2: Further accuracy information about inference intervention on Mistral-7B, Llama-7B, and Gemma-7B.

| Model | Method | Length | | | | | |
|---|---|---|---|---|---|---|---|
| | | 10 | 20 | 30 | 40 | 50 | 60 |
| Mistral-7B | Baseline (%) | 48.21 | 2.98 | 0.05 | 0.00 | 0.00 | 0.00 |
| | Inference Intervention (%) | 72.45 | 38.21 | 14.12 | 4.63 | 1.12 | 0.22 |
| Llama-7B | Baseline (%) | 0.04 | 0.00 | 0.00 | 0.00 | 0.00 | 0.00 |
| | Inference Intervention (%) | 3.73 | 3.51 | 0.06 | 0.00 | 0.00 | 0.00 |
| Gemma-7B | Baseline (%) | 7.73 | 0.08 | 0.01 | 0.00 | 0.00 | 0.00 |
| | Inference Intervention (%) | 15.45 | 4.32 | 1.14 | 0.42 | 0.11 | 0.00 |

The results show that the model has indeed developed two processing mechanisms. Ablation of the detection head and target head has no effect on $CC1$ but has a huge impact on more complex situations.

Table 3: Ablation study on the fine-tuned Gemma2-2B model (trained on a maximum length of 10).

| Method | $CC1$ | $CC4$ | $CC6$ | $CC10$ |
|---|---|---|---|---|
| Baseline | 100 | 99.67 | 99.62 | 91.72 |
| Detection Head Ablation | 100 | 55.42 | 58.37 | 54.32 |
| Target Head Ablation | 100 | 51.58 | 53.76 | 54.21 |
| Combined Ablation | 100 | 44.11 | 49.09 | 48.12 |
| Random Ablation | 99.68 | 99.12 | 99.72 | 90.47 |

# E    APPENDIX - IMPLEMENTATION DETAILS

The prompts in table 4 are applied in the fine-tuning experiment (randomly sampled), in the experiments related to model inference (Section 3, Section 4), the prompt is fixed to the first prompt shown in the table. The complete format is a prompt plus the question '$X + Y =$' format as input. The temperature is set as 1 and use greedy sampling. The reason for choosing accuracy instead of digit match as the evaluation metric is due to the accumulation of errors during model inference.

The detailed fine-tuning parameters are listed in table 5. The batch size (varies from 4 to 64) and taken epoch (usually 6-9) varies depending on the specific $CCd$ and $OCd$ tasks. The entire training process is done with one Nvidia A800 GPU, all experiments in the paper could be done within 15 hours.

The dataset includes questions of $CCd$, $OCd$, and randomly generated number pairs. To ensure that modular addition does not affect the results, 40% of the dataset consists of randomly sampled numbers with a length upper limit of 80 (sampled between 10 and $10^{79}$), 30% consists of $CCd$ tasks, and 30% consists of $OCd$ tasks. The dataset includes $10^6$ samples for each length $d$, which means $4 * 10^5$ randomly sampled numbers pairs, $3 * 10^5$ $CCd$ samples, and $3 * 10^5$ $OCd$ samples. For the $CCd$ and $OCd$ tasks, we add a number padding of length $d$ in the front and behind. For example, a $CC2$ input "234+468=702" could be "22344+34682=57026" in the dataset. The fine-tuning training process has musked the prompt, '$X + Y =$" sequence, and only predicts for $Z$.

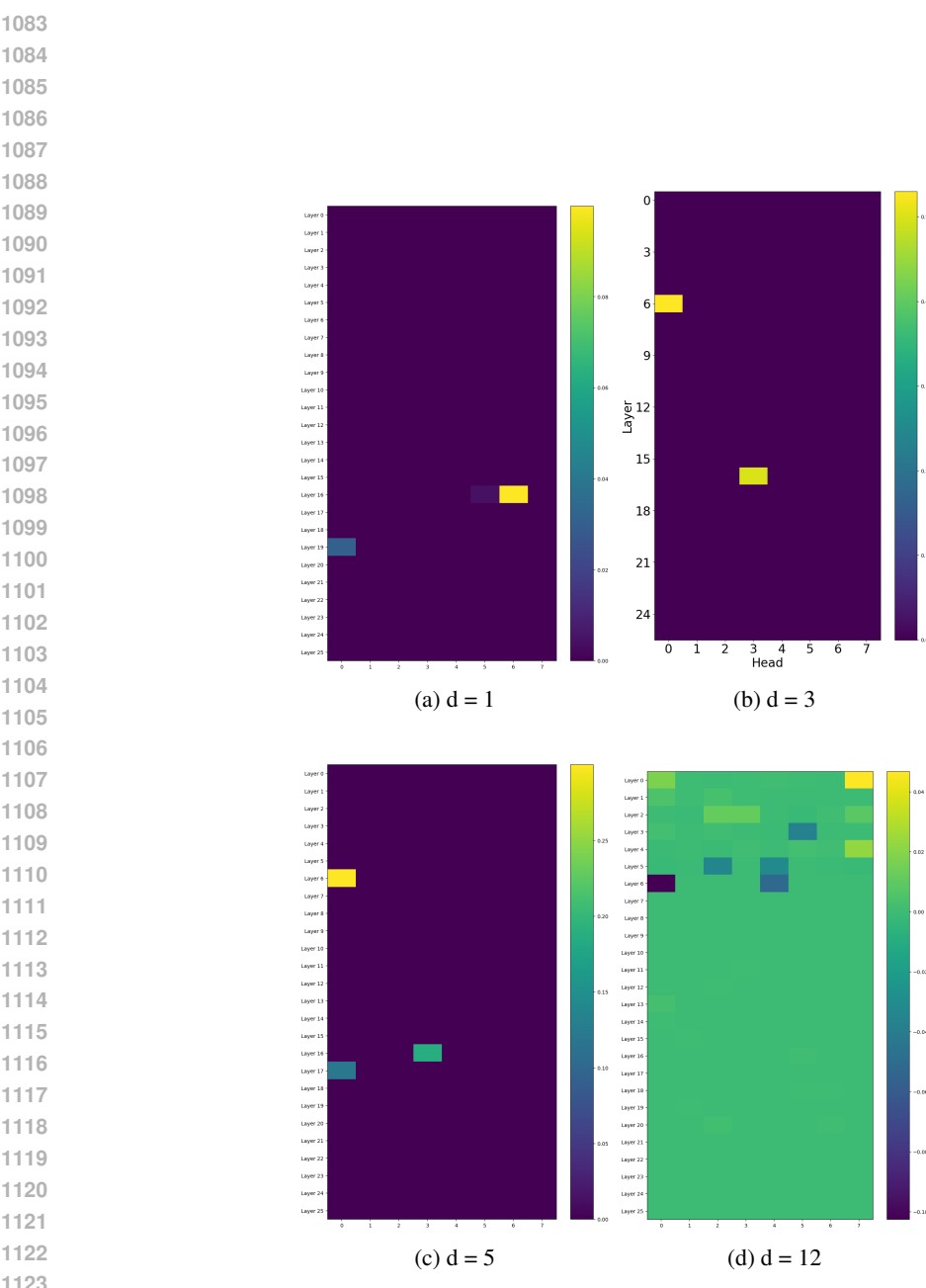

Figure 20: The attention heads located through causal analysis show that for fine-tuned Gemma2-2B under different values of $d$, $d = 12$ analysis is based on OOD input.

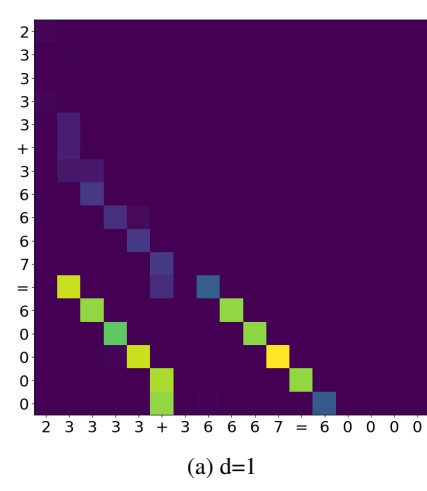

(a) d=1

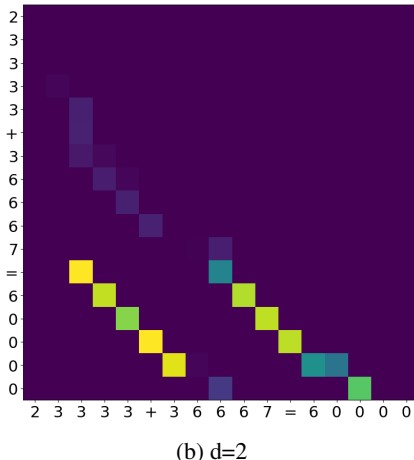

(b) d=2

Figure 21: The result of superimposing the attention heads on fine-tuned Gemma2-2B, 23333+36667=60000 is used as a demonstration example. For simplicity, the attention pattern displayed omits the prompt and only retains the question. For d from 1 to 2, the selected top 2 attention heads (Head, Layer) are (16, 6) and (19, 0); (16, 6) and (19, 9).

Table 4: Examples of Prompts

| **Prompt Examples** |
| --- |
| Do math calculations: |
| Calculate: |
| Compute the following sum: |
| Solve the addition: |
| Calculate the result of: |
| Solve the following problem: |
| Perform the calculation: |
| Determine the result of: |
| Find the value of: |
| Complete the calculation: |
| What is the solution to: |
| Solve this equation: |
| Compute the answer for: |
| What is the sum of: |
| Figure out the result of: |
| Determine the answer to: |
| Find the solution to: |
| Perform the operation: |

Table 5: Key Training Arguments Configuration, $d$ represents the $CCd$ and $OCd$ task.

| Parameter | Value |
|---|---|
| num_train_epochs | 15 |
| number of training tokens | about $(6d + 1) * 10^6$ |
| learning_rate | 5e-5 |
| bf16 | True |
| weight_decay | 0.0 |
| adam_beta1 | 0.9 |
| adam_beta2 | 0.999 |
| adam_epsilon | 1e-08 |
| gradient_accumulation_steps | 1 |
| seed | 42 |
| lr_scheduler_type | linear |
| optim | adamw_torch |

