# OpenReview forum: "IS TRANSFORMER A STOCHASTIC PARROT? A CASE STUDY IN SIMPLE ARITHMETIC TASK"
_ICLR.cc/2025/Conference — ICLR 2025 Conference Withdrawn Submission_

### Official Review · Reviewer_Yy1P · 2024-11-02

**Soundness:** 2
**Presentation:** 2
**Contribution:** 2
**Rating:** 1
**Confidence:** 5

**Summary:**

In this paper, the authors address the issue that large-scale language models based on Transformers often make mistakes in integer addition.
In integer addition, the result of the operation is often transferred from the lower digits to the higher digits. The authors define this transfer as the carry length and analyze it in detail.
The causal analysis showed that some attention heads strongly influence the output. (Sec 3.2)
Next, from the visualization of the attention pattern, it was found that the model predicts the next token by relying on a stair-shaped attention pattern, and when the carry becomes long, this stair-shaped pattern becomes disordered, and the value information is lost. (Sec 3.3)
From these analyses, the authors argue that many of the model's errors are due to incorrect processing of carries and propose a method for re-weighting the attention weights, and the experimental results show its effectiveness. (Sec 4)
In addition, the authors investigate how much the model can generalize to carry learning in out-of-distribution generalization. (Sec 5)

**Strengths:**

- This paper was written in an easy-to-understand way, so it was easy to understand. The theme is interesting.
- The proposed method also seems to be simple but effective.
- The experiment was conducted using three types of models (Mistral-7B, Llama-2-7B, Gemma-7B), so it is considered to be a comprehensive result.

**Weaknesses:**

- In the abstract of this paper, the authors claim that 'the formation of the attention heads is crucial to the length generalization,' but I cannot find any evidence in the main text to support this claim. Therefore, this claim should at least be removed from the abstract. (If the authors make this claim, they should discuss it with position encoding. Position encoding is the most critical factor in length generalization, and there has been much research on it. However, this paper does not mention RoPE, position interpolation methods, or extrapolation. )
- (L140-141)  h_i^0 = d_i + pos(i), this formula is incorrect because Mistral-7B uses RoPE. It would be better to rewrite it.
- There is no description of the fine-tuning parameter settings or detailed description of the data set. It may appear to be non-reproducible.

**I have repeatedly asked authors to correctly describe the model in their papers. However, they insist that they want the paper to be concise and that the description in the relevant section is not incorrect. Can't the authors even make this simple correction? Usually such mistakes are corrected perfectly once pointed out.
In the course of our discussion, I realized that they did not distinguish between the GPT model and the llama model. (I should have realized this possibility at the first stage of the description, e.g., d_i+pos.)
If they are writing an analytical paper on a model, the structure of the model should be described correctly. The authors were quick to respond, but only with excuses, never providing correct statements or rationales. I can teach them the correct description, but is that really the role of the reviewer? The quality of a paper is quite low when it is not correctly described in the first place.
This is a Transformer paper; the description of the Transformer model should be written correctly.**

**Papers that neglect description in favor of simplicity should not be accepted to top conferences.**

**As a supplementary note, I pointed out the issue of the position encoding in llama, but the author replied that they had referred to the GPT model paper[1]. If you know about the llama model and the GPT model, you should be able to tell that the position encoding is different. Therefore, it is wrong to refer to the GPT model paper to describe the position encoding of the llama model.**

**To add to this, the notation $pos_i$ is incorrect. This is proof that the author has not read the original papers on RoPE[2] and Transformer[3].In the original paper on Transformers, it is defined as $pos$, and in the original paper on RoPE, it is defined as $m$.  $pos$ expresses absolute position. In RoPE and Transformer papers, $i$ is not appended in $pos$. This is because absolute position is independent of $i$. Is it the reviewer's job to teach the author basic knowledge like this? I think that the author lacks basic knowledge. Because the paper written by such an author lacks discussion (both the author and the other reviewers admit that the discussion of position encoding is lacking), this paper is not worthy of acceptance.**


[1] Locating and editing factual associations in GPT, NeurIPS 2022

[2] Roformer: Enhanced transformer with rotary position embedding, Arxiv

[3] Attention Is All You Need, NeuriPS 2017

**Questions:**

1\. (Sec 3.3) There is a large variation in the baseline scores for CC4, CC6, and CC10. What is the cause of this?
In particular, the score for Llama2-7B is very low compared to Mistral-7B and Gemma-7B for CC10.

2\. (Sec 3.3) How long is the sequence length of CC10?
Looking at Figure 8, I think the sequence length of CC10 may exceed 4k.
If it does, the effect of max_position_embeddings is significant, so a discussion or explanation should be included.
The max_position_embeddings is 32k for Mistral-7B (window size is 4k), 4k for Llama2-7B, and 8k for Gemma-7B. Is this difference affecting the results in Table 1?

3\. (Sec 4.2) The experimental results are listed in Table 1, but it is difficult to understand because there is no citation in the main text. Please include a citation in Section 4.2.

---

> ### Author Response · Authors · 2024-11-19
>
> Thanks to careful reviews and comments from reviewer Yy1P
>
> **Weakness 1: In the abstract of this paper, the authors claim that 'the formation of the attention heads is crucial to the length generalization,' but I cannot find any evidence in the main text to support this claim. Therefore, this claim should at least be removed from the abstract. (If the authors make this claim, they should discuss it with position encoding. Position encoding is the most critical factor in length generalization, and there has been much research on it. However, this paper does not mention RoPE, position interpolation methods, or extrapolation. )**
>
> > **A1:** Thanks for raising the importance issue. This statement in our paper is an inappropriate expression. We think the statement: **The formation of the attention heads is crucial to the length generalization**  is a lack of rigor in writing (because it does not generalize well), we have modified the statement to: **This loss of generalization ability is related to the loss of ability to form functional attention heads**. While the statement has been removed from the abstract, we believe it remains a contribution of our work: to downgrade the generalization problem to the problem of attention formation. Future work can study on the model components related to it and further refine the problem to a specific level. (even though it may still return to the problem of position encoding in the end). To support this perspective, we have added Figure 8 (c) and the discussion in Section 5.1. And thanks for raising the issue about positional encoding, this is a must-discussed part when talking about the generalization on arithmetic tasks, which includes a discussion between positional encoding and carry generalization. We have added the discussion in Section 5.1. Additionally, we have added a new section 5.2 about the new ability model gains during fine-tuning, hope these additions would enhance the depth of the paper.
>
> **Weakness2: (L140-141) h_i^0 = d_i + pos(i), this formula is incorrect because Mistral-7B uses RoPE. It would be better to rewrite it.**
> >**A2:** Thanks for your careful reading. We have fixed the mistake and explained the formula more clearly.
>
> **Weakness3: There is no description of the fine-tuning parameter settings or detailed description of the data set. It may appear to be non-reproducible.**
> >**A3:** Thanks for raising the concerns. The details of training parameters and dataset descriptions are added in Appendix E.
>
> **Q1: (Sec 3.3) There is a large variation in the baseline scores for CC4, CC6, and CC10. What is the cause of this? In particular, the score for Llama2-7B is very low compared to Mistral-7B and Gemma-7B for CC10.**
> >**A4:** Thanks for constructive comments. We have added the explanation in Section 3.3 and details in Appendix C. The low accuracy of Llama2 can be attributed to its inherently lower performance in normal addition compared to the other two.
>
> **Q2: (Sec 3.3) How long is the sequence length of CC10? Looking at Figure 8, I think the sequence length of CC10 may exceed 4k. If it does, the effect of max_position_embeddings is significant, so a discussion or explanation should be included. The max_position_embeddings is 32k for Mistral-7B (window size is 4k), 4k for Llama2-7B, and 8k for Gemma-7B. Is this difference affecting the results in Table 1?:**
>
> >**A5:** Thanks for raising the question, the sequence length of CC10 is below 100, since the input sequence includes three numbers(X,Y, and Z) with each number length of at least 11, we extend the length to 20 by adding number padding (no carry inside only modular addition), so the sequence length of CC10 is (3*20+len(prompt)), which is far below the window size.
>
> **Q3: (Sec 4.2) The experimental results are listed in Table 1, but it is difficult to understand because there is no citation in the main text. Please include a citation in Section 4.2.**
>
> >**A6:** Thank you for pointing out this issue. Table 1 is indeed referenced in Section 3.3. We have now explicitly added a reference to the table in Section 4.2 along with a discussion on the results of inference intervention. Additionally, we have highlighted several key data points in bold for clarity. The question regarding Llama2 has also been addressed in this section. We hope these changes provide the reviewer with a clearer understanding.

---

> > ### Comment · Reviewer_Yy1P · 2024-11-23
> > **Reply from Reviewer Yy1P**
> >
> > Thank you for your reply. I have read everything.
> > Unfortunately, I will be lowering my score.
> > This is mainly because I looked at your corrections and decided that the quality of the paper is low for a large-scale language model paper, and that this paper is not at the level of the main conference.
> >
> >
> > The reasons are as follows:
> >
> > - The description of position encoding is still unclear. I checked the corrected formula at L115. I have read many papers on position encoding, but I have not seen this kind of expression before. Is this description correct? Are there any other papers that use this kind of expression? Also, it looks like the definition of l is not given. Papers with unclear or incorrect descriptions on the analysis of large-scale language models should not be accepted.
> >
> > - The discussion of position encoding is insufficient. Thank you for adding the discussion of position encoding to the beginning of section 5.1. However, when reading the main text, shouldn't you also introduce position encoding in the related research?
> >
> > - There are multiple prompts, but this is unclear. At first, I thought you were only using one prompt. In the verification of application tasks using LLM, in most cases, the prompt used is fixed to one. Is it normal to use multiple prompts in this task?
> >
> > - The dataset is unclear. At first, I thought you were using some famous dataset. However, you are fine-tuning using your original dataset. If you are using your original dataset, you should state the number of samples.
> >
> >
> > I would like to receive a response to my questions and doubts. I have doubts about the writing, but I think the theme of this research itself is interesting. (If I am wrong or have misunderstood something, please let me know!)
> > Of course, I will revise my score if my doubts are resolved. I look forward to your response.

---

> ### Author Response · Authors · 2024-11-23
>
> Thanks to reviewer Yy1P for your responses and we feel upset to see the negative feedback about the writing problems of details. We think that some issues do exist, but there are still some important issues that we think are inappropriate.
>
> **Reason1: The description of position encoding is still unclear. I checked the corrected formula at L115. I have read many papers on position encoding, but I have not seen this kind of expression before. Is this description correct? Are there any other papers that use this kind of expression? Also, it looks like the definition of l is not given. Papers with unclear or incorrect descriptions on the analysis of large-scale language models should not be accepted.**
>
> >**A1:** We agree that a detailed description is essential for ensuring soundness, but we also firmly argue that the level of detail should align with the scope of the research conducted. The use of the general operator $f(\cdot)$ in our paper does not, in our view, result in any lack of clarity or lead to misunderstandings. While our paper discussed the role of positional encoding in generalization, it is not the focus of our research, and we have not conducted any comparative or intervention experiments on positional encoding (as such study would require training models from scratch, which falls outside the scope of this work), as well as provide any insights or suggestion on it. Giving more details about positional encodings does not help the experiments and conclusions of the paper.
> That said, we have replaced the general operator $f(\cdot)$ with $RoPE(\cdot)$, as the models used in our experiments share the same positional encoding. Although the use of a general operator is uncommon, the meaning of the formula maintains, we have indeed seen it once in a paper related to interpretability (we have not yet been able to retrieve this reference). But considering the impact of positional encoding on generalization, we have added information about the limitation of RoPE in Section 5.1.  Lastly, this is an oversight of the definition of $l$ and have now added it. Thanks for pointing out.
>
> **Reason2: The discussion of position encoding is insufficient. Thank you for adding the discussion of position encoding to the beginning of section 5.1. However, when reading the main text, shouldn't you also introduce position encoding in the related research?**
>
> >**A2:** Thanks for the concerns. Again, this is a careful trade-off decision. Our experiments do not involve any experiments on position encoding, but entirely about model interpretability on arithmetic tasks, and the discussion part in Section 5.1 is enough for readers to reach out for more domain-specific studies. Similarly, we did not add the related studies on activation intervention to the related work but added it inside section 4. Adding it to the related work may lead readers to look at our work with different expectations (maybe a discovery on positional encoding, which is not the case).
>
> **Reason3: There are multiple prompts, but this is unclear. At first, I thought you were only using one prompt. In the verification of application tasks using LLM, in most cases, the prompt used is fixed to one. Is it normal to use multiple prompts in this task?**
>
> >**A3:** Thanks for raising the question, the use of multiple prompts is normal. In Study [1], which focuses on arithmetic fine-tuning, they utilized prompt constructions that are quite similar to ours during the fine-tuning process. During the model inference stage, currently lack a dedicated benchmark for evaluating performance in these exploratory experiments, but building such a benchmark is beyond the scope of this work. We are temporarily using the fixed prompt for inference (have identified in table 4). In future work, we will make a more comprehensive benchmark.
>
> **Reason4: The dataset is unclear. At first, I thought you were using some famous dataset. However, you are fine-tuning using your original dataset. If you are using your original dataset, you should state the number of samples.**
>
> >**A4:** Thanks for raising the question. After getting the first and second responses from the reviewer, we feel that there are some misunderstandings, so we would like to make some clarifications before answering the questions. Our work is quite exploratory (complex cases in addition tasks). As far as we know, this is the first time to specifically study this complex operation in arithmetic tasks (but fundamental to addition and arithmetic tasks). There is currently no dedicated benchmark dataset to evaluate or fine-tune our research object. Back to the reviewer's question, there is a description of a number of training tokens (about $(6d+1)*10^6 $) for each length of $d$ in Appendix E, to make this more clear, the dataset includes $10^6$ samples for each length of $d$, which we have added in Appendix E, thanks for point this out.
>
> **Reference**
>
> [1] Goat: Fine-tuned LLaMA Outperforms GPT-4 on Arithmetic Tasks

---

> ### Author Response · Authors · 2024-11-23
>
> Dear reviewer, we would like to add some additional thoughts and content just for your information:
>
> **Issues about the generalization and positional encoding**
> > We appreciate the reviewer's attention to positional encoding, but we have to make a clarification that the contribution of our study has little to do with the study on positional encoding. Although we did not focus and conduct experiments on positional encoding, we believe that further research on positional encoding and generalization of addition length is promising. Common arithmetic length generalization studies focus on normal number operations. In our study, a special part of our work is that this is a problem of a generalization within a generalization. Since the input number $X,Y$ length is greater than the
> length of $CCd$ or $OCd$  embedded inside, studies with a finer grain could have a positive impact on interpretability.
>
> **Issues about the writing problems of details**
> > We admit that some details are undoubtedly needed, while some need a trade-off. Hope that the information we added would help for your reading. We wish to have a further discussion with you

---

> ### Author Response · Authors · 2024-11-25
>
> >Dear Reviewer Yy1P,
>
> >Thank you once again for your constructive comments. We hope our response can resolve your concerns and hope to hear from you again. While we understand that reviewers have many other commitments, as authors, we are still eager to continue this discussion, incorporate your feedback, and improve our paper. If you have any further concerns, we are more than willing to provide additional clarifications or materials to ensure our work meets your expectations.

---

> ### Comment · Reviewer_Yy1P · 2024-11-25
>
> Thank you for your response. I understand reasons 2, 3, and 4. I was satisfied with reasons 2 and 3.
>
> Regarding question 1, (**I would like to emphasize that this is the third time I have drawn attention to the formula of Transformer**), it still seems to me that the correct description is lacking. There is no definition of $pos$ in the position encoding formula. Also, which paper has a similar formula?
> At first, I thought the equation would be corrected after one correction. However, even after two corrections, the equation in the paper was unclear.
>
> As the title says, this is a paper that analyzes Transformers. Therefore, a paper that does not correctly explain Transformers should not be accepted to the main conference.

---

> ### Author Response · Authors · 2024-11-25
>
> >Dear reviewer Yy1P,
>
> >Thanks for your quick response.
>
> >To answer your biggest concern. The similar brief expression of positional encoding expression can be found in [1], which is related to model interpretability. This is a study we have followed closely for a long time, and some of their expressions have influenced our own work. We have added the meaning of $pos(i)$. Since many symbols are already used in the paper, we just want to keep some simplicity. Hope you could understand.
>
>
>
> **Reference**
>
> Locating and editing factual associations in GPT.  NeurIPS 2022

---

> ### Author Response · Authors · 2024-11-25
>
> **Issues on the continued rebuttal on the expression of formula**
> >Dear reviewer Yy1P,
>
> >Thank you once again for your quick response. __We feel that the ongoing issue stems from differing expectations regarding the level of detail required for this specific problem, and we can not agree that an unusual format of a formula inherently makes it incorrect. Ultimately, we have decided to adopt the expression: $h_i=RoPE(d_i, pos_i, d_k)$__ (diff: more details added). We still don't know whether this is acceptable or not from your point of view. We think that no more details should be given since our paper conducts no study on it. Many papers address positional encoding within 10 words, while others—particularly those specializing in positional encoding—use highly detailed formulas. Neither of these extremes aligns with the intent of our work. If you still find this approach still unsatisfactory, we must ask for further clarification on why this expression might be problematic.

---

> ### Comment · Reviewer_Yy1P · 2024-11-26
>
> Thank you for your response to my comments. I checked the paper you told me about. They are using a GPT model. The positional encoding of GPT models is **not RoPE**. Therefore, the description used in the [1] paper should not be used as is.
>
> You may think that you are responding quickly and completely, but it seems that you don't even understand the structure of the model you are using. You should know the model you used in your experiments before writing a paper. At first, I thought that this might be a minor problem. **However, during this rebuttal period, it has become clear that you do not even understand the differences in the structure of the models you used in your experiments. If you understood the differences in the structure of the models, you would know that you should not have written model explanation with reference to [1].**
>
> I will not change the score.  **Is a paper that analyzes LLM without understanding the structure of the Transformer model suitable for the main conference? This is left to the judgment of the area chair. At least, I have never seen such a paper being accepted for the main conference**
>
> [1]Locating and editing factual associations in GPT. NeurIPS 2022

---

> ### Author Response · Authors · 2024-11-26
>
> Thanks for your response, we agree that it is time to stop the discussion and let the area chair decide.
>
> This is a tiny issue, but let's make the rebuttal process clear. Thank you for your first reminder on the mistake we made on it, so on the very first revision we have fixed it to $h_i = f(d_i,pos(i))$, __we use the general operator to represent the operation and the definition is explained in [2]__ (we have cited right after the formula), then we get a lower score and your response: 'The description of position encoding is still unclear.' We thought you might want the complete formula and think the $f(\cdot)$ is too abstract, but this is not suitable and no space at all for our paper to add extra formulas, even if full formulas were added, they would still be 'one-time formulas' that will never be mentioned in subsequent content, and no one does this when the paper does not study on it.
>
> So we changed to $h_i = RoPE(\cdot)$ to represent the operation more clearly, then we got your response again: 'It still seems to me that the correct description is lacking'. __The biggest thing we could not understand is that such a concise expression could have a problem__. At least, we asked many experts about the expression we use and none of them says it is problematic.
>
> At last, we thought that your saying 'correct description is lacking' is talking about the lacking variable inside the formula, because $d_k$ decides the frequency scaling ratio in different dimensions in RoPE, so we changed to $h_i = RoPE(d_i, pos_i, d_k)$ and cite the paper [1] just to illustrate that a concise and abstract formula can be used when details are not needed. And this time, you say that it's clear that we don't understand the model we have used.  __We also gradually understand why you refuse to accept the solution we provide because you are taking a skeptical perspective since the first revision to request us to meet your expectations on a non-domain-related paper.__ Our research has thoroughly examined the internal workings of the model, to perform our experiments, we have to study every part of these models, and we could never imagine that a model interpretability-related paper could receive such an unwarranted doubt on a tiny issue.
>
> From beginning to end, we continue to get your responses like 'still unclear', 'Are there any other papers that use this kind of expression', 'the correct description is lacking', 'Also, which paper has a similar formula?' __But we never get an explanation about why ours is problematic.__ Is it lacking correctness because it is abstract? Or is it lacking correctness because no one uses this expression? These two factors cannot be a proper reason. Your response is like a riddle for our authors.
>
> Many thanks for your other valuable comments, from a utilitarian perspective, we could include every detail of the formula in the revised version to meet your expectations. However, doing so brings no help and only makes the paper bloated. We have tried numerous times to explain this reasoning, but it seems our efforts only deepen your doubt.
>
>
>
> **References**
>
> [1] Locating and editing factual associations in GPT. NeurIPS 2022
>
> [2] Roformer: Enhanced transformer with rotary position embedding

---

### Official Review · Reviewer_SZsY · 2024-11-02

**Soundness:** 3
**Presentation:** 3
**Contribution:** 3
**Rating:** 6
**Confidence:** 4

**Summary:**

This paper offers a comprehensive and detailed investigation into how a Transformer performs simple addition. The authors support their findings by conducting inference interference to improve addition performance. Finally, they demonstrate that fine-tuning a model on the addition task does not significantly enhance its ability to perform addition.

**Strengths:**

- The experiments conducted in this paper are thorough.

- Exploring the limitations of Transformers to handle mathematical tasks is a crucial research focus, as it has significant implications for their use in quantitative applications.

**Weaknesses:**

- Additional details needed: What kind of positional embedding is used in the model? Previous research such as [1] has shown that positional embeddings play a crucial role. I am aware that [1] is cited, but the authors only briefly mentioned it sec 4.2.

- Fine-tuning: How many epochs were used by the authors to fine-tune the model? Can the grokking effect start to take effect with sufficient fine-tuning steps?

- Related work: The authors should position this paper more clearly within the related work section. Transformers with addition have been well-studied, and lines 90 to 93 mention at least two very similar studies. How does this work differ from them?

- Lack of an implementable algorithm: This paper does not provide an algorithm/circuit that can be directly implemented by a Transformer. That said, I understand the challenge of identifying a circuit for addition and do not expect the authors to provide one in the rebuttal.

[1] Transformers Can Do Arithmetic with the Right Embeddings

**Questions:**

See above.

---

> ### Author Response · Authors · 2024-11-19
>
> Thanks to reviewer SZsY for providing your valuable comments. Our responses to each of your concerns are shown below:
>
> **Q1: Additional details needed: What kind of positional embedding is used in the model? Previous research such as [1] has shown that positional embeddings play a crucial role. I am aware that [1] is cited, but the authors only briefly mentioned it sec 4.2.**
>
> > **A1:** Thanks for the constructive comments. We agree that positional encoding is vital and must be discussed in terms of length generalization, particularly in arithmetic tasks. We have added the discussion about positional encoding in Section 5.1.  And the information about positional encoding the pre-trained models implement (RoPE) is added Section 3.1
>
> **Q2:Fine-tuning: How many epochs were used by the authors to fine-tune the model? Can the grokking effect start to take effect with sufficient fine-tuning steps?**
>
> > **A2:** Thanks for raising the questions. The details of training parameters are added in Appendix E.  The epochs we have trained are 15 and use early stopping at about 6-9 epochs (depending on the task complexity). We have not observed any grokking effect yet as it may need much more training epochs. However, we do notice that a new mechanism emerges in the fine-tuned model, we have added the report in Section 5.2, hoping this would helps to your questions.
>
> **Q3:Related work: The authors should position this paper more clearly within the related work section. Transformers with addition have been well-studied, and lines 90 to 93 mention at least two very similar studies. How does this work differ from them?**
>
> > **A3:** Thanks for raising the important issue. We have added a clearer illustration of our work in related works. The difference and contribution in our work is providing a general and applicable method for analysis of how the model handles arithmetic task, and the findings we provide exist on a very wide scale.
>
> **Q4: Lack of an implementable algorithm: This paper does not provide an algorithm/circuit that can be directly implemented by a Transformer. That said, I understand the challenge of identifying a circuit for addition and do not expect the authors to provide one in the rebuttal.**
>
> > **A4:**  Thanks for your concerns about the some lacks of our work. Fingering out a certain circuit is indeed beneficial to the LLM interpretability community, we have reconsidered the contribution our works can bring and try to give more insightful findings. As we have added in Section 5.2, we believe some circuit studies could based on these findings because functional differentiation on different model components is crucial for circuit analysis.
>
> **Reference**
>
> [1] Transformers Can Do Arithmetic with the Right Embeddings

---

> > ### Comment · Reviewer_SZsY · 2024-11-21
> > **Thank you for the response**
> >
> > Dear authors,
> >
> > Thank you for your response. I will retain my original score and lean toward accepting this paper.

---

> > > ### Author Response · Authors · 2024-11-23
> > >
> > > Sure, thanks for the careful reviews from reviewer SZsY, and thanks for your approbation.

---

### Official Review · Reviewer_RWwi · 2024-11-04

**Soundness:** 3
**Presentation:** 1
**Contribution:** 2
**Rating:** 3
**Confidence:** 4

**Summary:**

The paper attempts to study the carry propagation error for the task of addition when LLMs such as mistral and Lllama2 are doing the calculations. Through some ablation study, they find the attentions heads responsible for this error, and make an intervention to offset this error.  Accordingly, they suggest fine-tuning the model based on this to get better results.

**Strengths:**

1- The paper has developed a natural story from doing the ablation study to fine-tuning a model in Sections 3 to 5, which makes it more understandable to a general reader.

2- To my knowledge, this is the first attempt for doing an ablation study for the task of addition in a model as large as llama2. Hopefully, this will contribute to the mechanistic interpretability of LLMs for logical tasks such as addition and parity.

**Weaknesses:**

1- Before, jumping to the details of experiments in section 3, it would’ve been more convenient for the reader if they had described the input and the output format. For instance, I was curious about the answer of Llama2 and prompted “233335 + 566667” twice for temperature = 1 and got the two answers below — While the final answer is the same, the format is different and needs some extra care. Did you read the final answer manually across all your experiments?  Did you set the temperature to zero? How do you ensure that the output format stays the same?

```
Sure! Here is the result of adding 233335 and 566667:
233335 + 566667 = 799992

Sure! Here is the calculation:
233335 + 566667 = 799992
```


2- In line 177, initially I thought it must be $x_{d+1} + y_{d+1}$ and $x_{d+1}' + y_{d+1}'$. I had to go over the entire section to understand that it is not the case and reason for that choice. The authors must have done a better job at explaining their experimental set up. Another example of this are equations (1) to (3) where attention scores and $m_i$’s are given without any explanation. For example, what are $\sigma$ and $\gamma$?

3- In line 200, $v$ is used for the first time in the paper. I’m still not sure what the definition of this entity is. Because of this I couldn’t understand the ablation study in section 3.3, which is the core idea of the entire work and is later used in sections 4 and 5.

4- Figure 6 is not being referred anywhere in the main body. While it is probably for the results of Section 4, please make sure to refer the reader to the right place for checking up the results. Also, this figure only includes the accuracies up to length of 20, whereas in 417 they talk about a case where sequences of length 60 are involved.

5- If the ablation study demonstrates the main cause for the carry propagation errors in a large model such as mistral, why is there no improvement in figure 6 for up to length 5? This must be discussed by the end of Section 4.2.

6- Can you compare your work to some previous result that has attempted to address the carry propagation issue? For instance, the paper below:

[1] Sabbaghi et al., Explicitly Encoding Structural Symmetry is Key to Length Generalization in Arithmetic Tasks.

7- What is the effect of fine-tuning on tasks unrelated to the addition? Does the explained procedure do any harm to the typical benchmark score of llama2 or mistral?

**Questions:**

Please address the concerns expressed above.

---

> ### Author Response · Authors · 2024-11-19
>
> Thanks to reviewer RWwi for the constructive comments, we do realize that some expression issues need to be fixed. Below, we provide our point-wise responses to each of your concerns:
>
> **Q1: Before, jumping to the details of experiments in section 3, it would’ve been more convenient for the reader if they had described the input and the output format. For instance, I was curious about the answer of Llama2 and prompted “233335 + 566667” twice for temperature = 1 and got the two answers below — While the final answer is the same, the format is different and needs some extra care. Did you read the final answer manually across all your experiments? Did you set the temperature to zero? How do you ensure that the output format stays the same?**
> > **A1:** Thanks for your constructive suggestions. The details of the format, prompt, decoding strategy, and temperature are listed in Appendix E. We set the temperature as 1 and use greedy sampling, the output would directly be the answer.
>
> **Q2: In line 177, initially I thought it must be $x_{d+1}+y_{d+1}$ and $x_{d+1}' + y_{d+1}' > 9$. I had to go over the entire section to understand that it is not the case and reason for that choice. The authors must have done a better job at explaining their experimental setup. Another example of this are equations (1) to (3) where attention scores and
> $m_i'$  are given without any explanation. For example, what are $\sigma$
>  and $\gamma$**
>
> > **A2:** Thanks for pointing this out, we agree that our experiment process and principles are not clear enough. We have added a new figure and explanations in Sections 3.1 and 3.2 to interpret the casual analysis process that general readers can also understand easily. The meaning of each symbol is clearly explained
>
> **Q3: In line 200, $v$ is used for the first time in the paper. I’m still not sure what the definition of this entity is. Because of this I couldn’t understand the ablation study in section 3.3, which is the core idea of the entire work and is later used in sections 4 and 5.**
> > **A3:**: Thanks for pointing out this important issue, sorry for the huge difficulty in understanding we caused, this is an urgent problem. We have rewritten the formula part carefully in Section 3.1. The $v$ is a model component in attention layers. We hope these revisions provide a smoother reading experience : )
>
> **Q4: Figure 6 is not being referred anywhere in the main body. While it is probably for the results of Section 4, please make sure to refer the reader to the right place for checking up the results. Also, this figure only includes the accuracies up to length of 20, whereas in 417 they talk about a case where sequences of length 60 are involved.**
> > **A4:**: Thanks for the careful review. We have added the reference in Section 4.2. The accuracy data of up to the length of 60 is also added in Appendix C
>
> **Q5: If the ablation study demonstrates the main cause for the carry propagation errors in a large model such as mistral, why is there no improvement in figure 6 for up to length 5? This must be discussed by the end of Section 4.2.**
> > **A5:**: Thanks for the feedback. We have added the discussion in Section 4.2. In simple terms, the reason why there are no improvements for the length of up to 5 is that the proportion of complex cases in these short sequences is low, while our method benefits most in complex cases
>
> **Q6: Can you compare your work to some previous result that has attempted to address the carry propagation issue? For instance, the paper [1]:**
> > **A6:** Thanks for providing the paper. It argues relative positional encoding improves the ability to generalize addition tasks, the concept of 'cascading carries' in their paper corresponds to $CCd$ task in ours, the shared finding is that 'cascading carries' cases is still challenging(even for the model with Relative Positional Encoding), as it does not generalize well in out-of-distribution data. We have added the citation in Section 5.1 where positional encoding is discussed.
>
> **Q7: What is the effect of fine-tuning on tasks unrelated to the addition? Does the explained procedure do any harm to the typical benchmark score of llama2 or mistral?**
> > **A7:** Thanks for the question. The fine-tuning will have some negative effects to some extent for sure, but we'd like to clarify that the purpose of the fine-tuning part is to investigate the model's ability to generalize, and the new capabilities of the model (a new part we have added in the fine-tuning part, which we believe could provide you valuable insights). One of the direct suggestions our paper provides is to explicitly add complex cases in the training process(forming carry mechanism,also mentioned in [1]). The study of the benchmark on unrelated tasks could be done in further studies (training strategy to improve the performance of arithmetic task)
>
> **Reference**
>
> [1] Sabbaghi et al., Explicitly Encoding Structural Symmetry is Key to Length Generalization in Arithmetic Tasks.

---

> > ### Author Response · Authors · 2024-11-30
> >
> > Dear Reviewer RWwi,
> >
> > Thank you again for your comments. We truly value your feedback and hope to receive your thoughts on our revisions, we have made several revisions to address your concerns. Specifically, to resolve the difficulties you mentioned in understanding the experimental process, we have reformatted Sections 2 and 3. Every formula symbol is now clearly defined, and we have added an intuitive figure to better illustrate our experiments. We are confident that these changes will help clarify the study for you. Additionally, we have included the necessary experimental details as requested.
> >
> > If you have any further issues to raise, we are more than willing to address them.

---

> ### Author Response · Authors · 2024-11-25
>
> >Thanks to the reviewer RWwi for reviewing our paper and providing valuable feedback. We have carefully studied your concerns about the issues with unclear expressions and made multiple revisions to enhance the clarity and contributions of the paper.
>
> >Hoping that these revisions and clarifications will encourage you to reassess your evaluation, as the updates directly address your constructive comments. If there are any other questions or concerns, we are more than willing to provide additional clarifications or supporting materials.
>
> >We sincerely hope you will continue to engage in the discussion. If you have further questions or concerns, we are more than willing to provide additional clarifications or supporting materials!

---

> ### Comment · Area_Chair_LGo7 · 2024-11-25
>
> Dear Reviewer RWwi,
>
> Thank you for your valuable contributions to the review process for the paper! The authors have submitted their rebuttal, and I would greatly appreciate it if you could take a look and provide your response.

---

### Official Review · Reviewer_XD7d · 2024-11-05

**Soundness:** 3
**Presentation:** 3
**Contribution:** 3
**Rating:** 6
**Confidence:** 3

**Summary:**

This paper studies how pre-trained autoregressive language models perform addition operations. It founds that the model relies on localized attention distribution for handling carry operations. Thus it's hard for the model to process inputs with long sequences. The paper tries to restore the model's performance by intervening in attention during inference without training. The experimental findings show that autoregressive models rely on the staircase attention patterns to transmit carry-encoded value tokens to the final token for prediction Since their analysis shows that many errors stemmed from incorrect handling of carries, they try to restore the model's internal mechanisms by modifying the attention weights depending on whether there is a carry or not. The paper also tried fine-tuning with long addition data, with carry length up to 10. While fine-tuning is effective in improving the accuracy, it doesn't extend the model's ability to do addition with correct carries beyond 10, the limit set in the training time.

**Strengths:**

- Carefully designed tasks and intervention techniques to understand how the model performs addition. From the analysis, designing data to test the models' abilities to do addition with long carry chain.

**Weaknesses:**

- Finetuning experiments are done with one of the pre-trained models in the analysis.

**Questions:**

- Have the authors considered similar experiments for substraction?
- How does it affect the model's performance if we revert the input so that it becomes the order that humans perform addition?
- What's the authors thought on whether the model are merely statistical or there is deeper reasoning mechanism embedded in the model.

---

> ### Author Response · Authors · 2024-11-19
>
> Thanks to the comments from reviewer XD7d. Below, we provide our responses to each of your concerns:
>
> **Weakness: Finetuning experiments are done with one of the pre-trained models in the analysis.**
>
> > **A1:** Thanks for providing the concern. We are very willing to conduct fine-tuning experiments on all the models we have studied. Unfortunately, our budget is limited. In fact, this is also the reason why we chose gemma2-2B as the fine-tuning object. Our equipment is not enough to support the fine-tuning of the 7B model, and llama and Mistral do not provide 2B models. Additionally, the constraint of the tokenizer has further reduced our choices. In the end, we can only choose gemma2-2B. Hoping reviewer can understand our limited conditions.
>
> **Question: Have the authors considered similar experiments for substraction?**
> > **A2:** Yes, the entire experiment setup can be copied to the substraction task. Figure 6 shows that the ablation based on addition also work on substraction. The deeper reason is that addition and substraction share the same functional attention heads, which is not surprising (there is no essential difference between subtraction and addition).
>
> **Question: How does it affect the model's performance if we revert the input so that it becomes the order that humans perform addition?**
> > **A3:** Thanks for the insightful question. The question is well discussed in study [1]. Models generally have better performance when the input format is reverted (training needed). We have cited it in related works.
>
> **Question: What's the authors thought on whether the model are merely statistical or there is deeper reasoning mechanism embedded in the model.**
> > **A4:** Thanks for the very important question. We realize that we need to be careful with this issue so much that we modified some expressions in our paper. We hope reviewer could have a reading on Section 5.2 (a new section we have added, which is valuable to the stochastic parrot question). At present, our view is neutral. Although pre-trained models seem to use fixed attention patterns to achieve addition, a new mechanism that involves sum judgment emerges after fine-tuning and works together with the old mechanism.
>
> **Reference:**
>
> [1] Lee, Nayoung, et al. "Teaching arithmetic to small transformers."

---

> > ### Comment · Reviewer_XD7d · 2024-11-25
> >
> > I would like to thank the authors for their reply and I would keep my score, learning toward accepting the paper.

---

### Note · Authors · 2025-01-23

I have read and agree with the venue's withdrawal policy on behalf of myself and my co-authors.